# D-FUSEr: Diverse Failure, Unified Success
# via Error-Distribution Shaping in LLM Reasoning

**David D. Baek** [* 1 2]  **Andrew Estornell** [* 3]  **Yichi Zhang** [* 4]  **Muhammad Faaiz Taufiq** [3]
**Jean-Francois Ton** [3]  **Jie Mei** [5]  **Tao Wang** [5]

## Abstract

Test-time scaling methods such as majority vote aggregation and iterative refinement (e.g., self-reflection or multi-agent inference) improve reasoning performance by leveraging multiple solution samples. However, their efficacy depends not only on raw performance, but critically on the distribution of errors across samples. When errors concentrate, (a) aggregation accuracy degrades, as the majority vote may select a shared mistake, and (b) confidence in common mistakes may suppress exploration in iterative refinement. We argue that improving correctness alone is not sufficient to mitigate these issues; to this end, we propose to explicitly shape error distributions to improve aggregation. First, we introduce a theoretically grounded **diverse failure reward** that incentivizes calibrated disagreement within model errors. We prove that this reward directly optimizes majority-vote accuracy: policies achieving higher reward attain higher expected majority-vote performance, and vice versa. We further show that this theoretical property generalizes to iterative refinement. Second, we introduce **anti-votes**, in which the model predicts the most common mistake alongside its solution, allowing probability mass on dominant errors to be reweighted. We identify conditions under which anti-votes are guaranteed to improve majority-vote accuracy. Empirically, across three model families of varying sizes and four benchmarks, we show that both approaches substantially improve majority vote and iterative refinement performance without degrading single-sample accuracy.

[1]Massachusetts Institute of Technology, Cambridge, MA, USA [2]Work initiated during internship at ByteDance USA. [3]ByteDance Seed [4]Rutgers University, Newark, NJ, USA [5]ByteDance. Correspondence to: David Baek <dbaek@mit.edu>, Andrew Estornell <andrew.estornell@bytedance.com>.

*Proceedings of the $43^{rd}$ International Conference on Machine Learning*, Seoul, South Korea. PMLR 306, 2026. Copyright 2026 by the author(s).

## 1. Introduction

Improving LLM reasoning typically relies on two complementary levers: *training*, which shapes how models generate solutions, and *inference-time scaling*, which uses extra compute at test time to extract better answers. Inference-time scaling includes methods such as chain-of-thought prompting (Wei et al., 2022), majority vote (Wang et al., 2023), best-of-N sampling (Gui et al., 2024), iterative refinement (e.g., backtracking or self-reflection) (Madaan et al., 2023), and budget forcing (Muennighoff et al., 2025).

Among these, two widely used tools are (i) *majority vote*, which aggregates several sampled answers into one prediction, and (ii) *iterative refinement*, where the model updates its answer after seeing additional context. The success of these two methods depends on a factor that is easy to overlook, namely the *distribution of errors* across samples. As exemplified in Figure 2, if many samples repeat the same mistake, majority vote can select that shared error, and refinement can become biased toward it if not properly guided towards exploring lower density alternative solutions. Such behavior is not a corner case and was observed in several prior works (Wolfram & Schein, 2025; Zhang et al., 2025; Feng et al., 2025). In many settings, model errors are indeed highly concentrated (see Figure 3) and frequency often becomes a proxy for correctness during inference-time scaling, leading to diminishing returns from additional sampling.

This perspective suggests that correctness alone is an insufficient training target when the goal is to improve inference-time aggregation, such as majority vote or iterative refinement. What matters is not only whether an individual sample is correct, but how errors are distributed across samples when the model is wrong. Majority vote is most effective when the correct answer is stable and incorrect answers are dispersed. Similarly, iterative refinement benefits when trajectories continue to explore alternative hypotheses, rather than collapsing early onto a single incorrect solution that dominates subsequent updates. These observations point to a common underlying objective to pursue: shaping the error distribution induced by the model's policy.

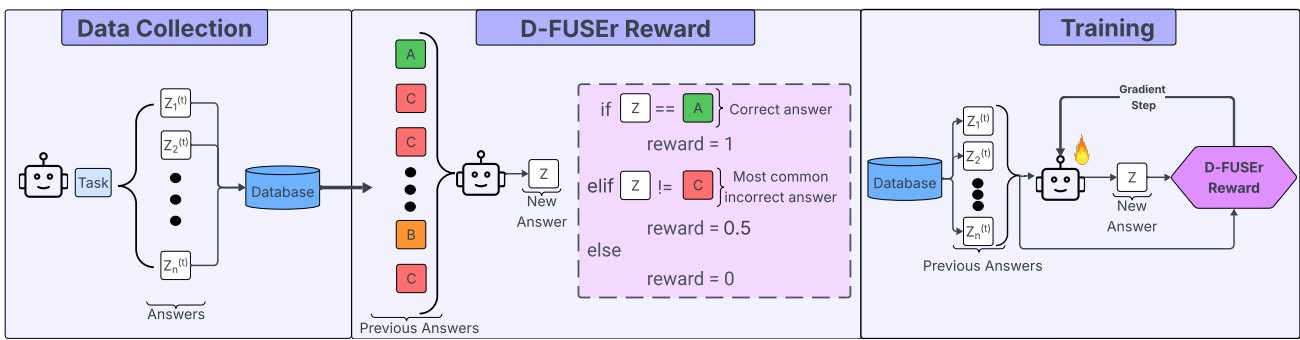

*Figure 1.* Outline of D-FUSEr framework. (1) **Data Collection:** We generate training data by sampling responses using off-the-shelf models. The target model is tasked with generating a refined response conditioned on the distribution of these responses. (2) **D-FUSEr Reward:** Our reward function consists of the correctness reward, and failure reward. The failure reward is applied only when the model's answer is incorrect, and differs from the most common incorrect answer in the previous round. (3) **Training:** With the collected data and the defined reward function, any reinforcement learning framework can be applied to train the models. We also introduce anti-votes, in which the model predicts the most likely incorrect answer alongside its solution, which is not illustrated in this figure.

**Contributions:** In this paper, we propose **D**iverse **F**ailure **U**nified **S**uccess via **Er**ror Distribution Shaping (**D-FUSEr**), a unified framework for improving inference-time scaling by explicitly controlling how errors are distributed across samples. The starting point of D-FUSEr is *distributional reflection*, a refinement paradigm in which self-reflection is conditioned on the distribution of previously sampled candidates, rather than on a single prior response. Distributional reflection prevents self-reinforcing errors by allowing the model to reason about rare answers nearly as much as more common ones. This capability is especially important in the presence of error concentration, where most probability mass is assigned to a small set of distinct but incorrect answers. In such settings, conventional self-reflection often fails, as refinement steps amplify the same error rather than correcting it. Building on this perspective, D-FUSEr provides two complementary mechanisms for making effective use of distributional reflection:

First, we introduce a theoretically grounded training objective, the **diverse failure reward** $r_{\text{D-FUSEr}}$, which encourages calibrated and diverse errors while preserving single-sample accuracy. We show that optimizing this reward directly improves expected majority-vote accuracy, both in the zero-shot setting and when combined with iterative refinement, by reducing the dominance of correlated failures.

Second, we propose **anti-votes**, an inference-time mechanism in which the model explicitly predicts likely incorrect answers. Anti-votes enable direct down-weighting of dominant errors during aggregation, further mitigating error concentration. We identify sufficient conditions under which anti-votes strictly improve majority-vote accuracy, and confirm empirically that they yield consistent gains in practice. Moreover, we provide a theoretical guideline on how to optimally weight anti-votes with the original votes.

Across three model families (Qwen-2.5, Llama-3, and Gemma-2), spanning sizes from 2B to 14B, and multiple benchmarks (MMLU-Pro, BBH, GPQA, and RBENCH), we demonstrate that combining diverse failure reward or anti-votes with distributional reflection improves both majority vote and iterative refinement without, including on unseen benchmarks. We also analyze the empirical structure of model errors and show that error distributions are often highly concentrated, with most probability mass assigned to a small number of distinct incorrect answers. This analysis highlights the necessity of error distribution shaping.

## 2. Preliminaries

Consider a task instance $x$ with $m$ possible answers. A policy $\pi_\theta$ generates a textual response $o \sim \pi_\theta(\cdot \mid x)$, from which a discrete answer is extracted:

$$Z := \text{ans}(o) \in [m] := \{1, \dots, m\}.$$

Let $y \in [m]$ be the correct answer.

**Majority vote aggregation.** Given $n$ independent sampled textual responses $o_i, \dots, o_n \sim \pi_\theta(\cdot \mid x)$ with answers $Z_i := \text{ans}(o_i)$, let $\vec{N} = (N_1, \dots, N_m)$ denote the vote counts, where $N_j$ is the number of times answer $j$ appears and $\sum_j^m N_j = n$. The majority-vote outcome is

$$Z_{\text{maj}} := \arg\max_{j \in [m]} N_j,$$

**Iterative refinement.** Consider a $T$-step refinement process; at round $t + 1$, the policy conditions on the task $x$ and a refinement context $C^{(t)}$ derived from previous outputs:

$$o_i^{(t+1)} \sim \pi_\theta(\cdot \mid x, C^{(t)}), \quad t \in [T], \text{ and } i \in [n],$$

with $C^{(1)}$ empty. At each round, an answer $Z_i^{(t)} := \text{ans}(o_i^{(t)})$ is extracted, and aggregation may be applied to samples from the final round. The refinement context is method-specific and defined in subsequent sections.

# 3. Methodology

In this section, we describe the methodology of our framework, **D-FUSEr** (**D**iverse **F**ailure **U**nified **S**uccess via **Er**ror Distribution Shaping). First, we introduce *distributional reflection*, an inference-time refinement paradigm that conditions updates on the distribution of previously sampled solutions. Second, we propose two mechanisms for shaping error distributions: **diverse failure reward** training and **anti-votes**. Finally, we describe the details of our training.

## 3.1. Distributional Reflection

Standard self-refinement methods condition each update on a single previous response or its critique. While effective in some settings, this approach can fail when the model repeatedly produces the same incorrect answer, causing refinement trajectories to collapse. To address this limitation, we introduce *distributional reflection*, which conditions updates on the distribution of solutions produced by the model.

Let $\tilde{p}^{(t)} \in \Delta^m$ denote the empirical distribution of the answers $Z_i^{(t)}{}_{i=1}^{n}$ at round $t \geq 1$, where $\Delta^m$ denotes the $m$-dimensional probability simplex.

$$\tilde{p}_j^{(t)} := \frac{1}{n} \sum_{i=1}^{n} \mathbf{1}\{Z_i^{(t)} = j\}.$$

Distributional reflection generates a new response by conditioning on this empirical answer distribution:

$$o_i^{(t+1)} \sim \pi_\theta \left( \cdot \mid x, \tilde{p}^{(t)} \right), \qquad Z_i^{(t+1)} := \text{ans}(o_i^{(t+1)}).$$

This allows the model to reason about which answers are common, which are rare, and how concentrated its uncertainty is, rather than reacting only to a single prior attempt.

In practice, the distribution $\tilde{p}^{(t)}$ is summarized into a compact context $C_{\text{dist}}^{(t)}(x)$. In our experiments, this context consists of the empirical distribution over answer choices (for example, the number of times each option is selected), along with one representative full response for each answer choice. We provide an example prompt in Appendix G.

**Relationship with Multi-agent Debate.** Like multi-agent debate (Du et al., 2023), distributional reflection exposes a model to multiple solutions. However, debate conditions on a small set of individual responses (typically from different models), without encoding their relative frequency or dominance, and thus does not reflect the underlying solution distribution. In contrast, distributional reflection conditions explicitly on an approximation of the model's own prior distribution, making errors and uncertainty directly visible.

## 3.2. Error Concentration

A key challenge for both majority vote and distributional reflection is *error concentration*. When a model is incorrect,

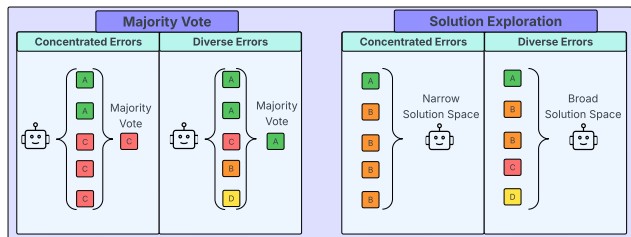

*Figure 2.* Illustration of how error diversity affects inference-time scaling. **Left:** When errors are concentrated, majority vote selects an incorrect answer. When errors are diverse, incorrect vote mass is dispersed and the correct answer is more likely to prevail. The key intuition is that while there are only a few ways to solve a problem correctly, there are many ways to make mistakes. **Right:** Greater error diversity also encourages solution space exploration without sacrificing the correctness. This can be especially helpful when one of the minority answers is the correct answer.

it often assigns most probability mass to a small number of distinct wrong answers, as we empirically verify in Figure 3. In this regime, the majority vote tends to select the common mistake, and refinement may reinforce it. We define the empirical majority answer as (omitting the round number) $\tilde{Z}_{\text{maj}} := \arg\max_{j \in [m]} \tilde{p}_j$, and the *majority error mode* as

$$\tilde{Z}_{\text{err}} := \arg\max_{j \in [m] \setminus \{y\}} \tilde{p}_j,$$

with ties broken uniformly at random. If all samples are correct, $\tilde{Z}_{\text{err}}$ is undefined. Our goal is to reduce error concentration by shaping the distribution of incorrect answers while preserving average correctness.

## 3.3. Diverse Failure Reward

We first introduce an implicit mechanism for shaping error distributions via training. For a sampled answer $Z$, we define **Diverse Failure Reward** as

$$r_{\text{D-FUSEr}}(Z; y, \tilde{Z}_{\text{err}}) := \mathbf{1}\{Z \notin \{y, \tilde{Z}_{\text{err}}\}\} \cdot \mathbf{1}\{\tilde{Z}_{\text{err}} \text{ is defined}\}.$$

Intuitively, when the model is wrong, it is rewarded for being wrong in a way that differs from the most common incorrect answer. The total reward is given by

$$R(Z; y, \tilde{Z}_{\text{err}}) := r_{\text{corr}}(Z; y) + 1/2\, r_{\text{D-FUSEr}}(Z; y, \tilde{Z}_{\text{err}}),$$

where $r_{\text{corr}}(Z; y) := \mathbf{1}\{Z = y\}$, In Section 4, we show that optimizing the expected D-FUSEr reward improves expected majority-vote accuracy and reduces collapse during refinement. Our reward design can be used with any reward-based training method. In our experiments, we train with Dr. GRPO (Liu et al., 2025b) for its strong performance.

## 3.4. Anti-votes

While the reward shapes errors implicitly through training, anti-votes provide an explicit inference-time mechanism for

down-weighting shared mistakes. In addition to an answer, each sampled response may include an anti-vote $A_i \in [m]$, indicating a prediction of the common mistake. Given $n$ samples, we define the empirical anti-vote distribution

$$\tilde{q}_j := \frac{1}{n} \sum_{i=1}^n \mathbf{1}\{A_i = j\}.$$

Note that anti-votes are also compatible with iterative refinement, while we omit the superscript $t$ here for simplicity. We then compute a reweighted score

$$S_j := \tilde{p}_j - \eta \, \tilde{q}_j$$

where $\eta \geq 0$ controls the strength of the anti-vote contribution. The final prediction is given by $Z_{\text{anti-maj}} := \arg\max_{j \in [m]} S_j$.

## 4. Theory

In this section, we provide a set of theoretical results/guarantees for our methods. We first characterize when one error distribution yields higher majority-vote accuracy than another, then show that our reward $r_{\text{D-FUSEr}}$ steers errors in this direction. We further give sufficient conditions under which this guarantee extends to iterative refinement, and conclude by discussing when anti-votes are strictly beneficial and what properties of an anti-vote distribution are preferred. We defer all proofs to Appendix B.

We start by introducing some global notations. Given two LLMs' answer distributions $p$ and $p'$, suppose without loss of generality that the correct answer is $y = 1$ and $p_j, p'_j > 0$ for any $j \in [m]$. Let $e = (p_2, \ldots, p_m)/(1 - p_1)$ and $e' = (p'_2, \ldots, p'_m)/(1 - p'_1)$ be the error distributions. Let $\text{Acc}^{\text{MV}}(p; n)$ be the majority-vote accuracy with $n$ i.i.d./ samples from distribution $n$.

### 4.1. Error Entropy and Majority-vote Accuracy

**Definition 4.1 (Majorization).** Let $v, v' \in \mathbb{R}^m$ be two vectors, and let $v^\downarrow$ and $v'^\downarrow$ denote the vectors obtained by sorting the entries of $v$ and $v'$ in non-increasing order. Then, $v$ weakly majorizes $v'$, i.e., $v \succeq v'$, if $\sum_i^k v_i^\downarrow \geq \sum_i^k v_i'^\downarrow$ for any $k \in [m]$. Moreover, $v$ strictly majorizes $v'$, i.e., $v \succ v'$, if the above inequality is strict for at least one $k$.

**Definition 4.2 (Error Entropy).** We say an error distribution $e$ has a higher error entropy than $e'$ if $e \succeq e'$.

Next, we show that higher error entropy leads to lower majority-vote accuracy.

**Proposition 4.3 (Higher Error Entropy Lowers Majority Accuracy).** *Given two answer distributions $p, p' \in \Delta^m$ with $p_1 = p'_1 \in (0,1)$, if $e' \succeq e$, then $\text{Acc}^{\text{MV}}(p; n) \geq \text{Acc}^{\text{MV}}(p'; n)$ for $n \geq 1$.*

**Corollary 4.4 (Uniform Error Maximizes Majority Accuracy).** *Fixing $p_1$, the uniform error distribution maximizes the majority-vote accuracy.*

Corollary 4.4 follows directly from Proposition 4.3 as the uniform error distribution has a strictly lower error entropy than any other error distribution $e \in \Delta^{m-1}$.

### 4.2. Error Entropy and Reward

Now, we relate error entropy to the proposed reward function. Consider an LLM with answer distribution $p \in \Delta^m$. The reward function draws $\bar{n}$ i.i.d./ reference samples from $p$ to identify the majority mistake $Z_{\text{err}} \in [m]\backslash\{y\}$, with uniform tie-breaking. Given a new answer $Z \sim p$, the total reward is $R(Z; y, Z_{\text{err}}) = \mathbb{I}[Z = y] + \frac{1}{2} \cdot \mathbb{I}\{Z \neq Z_{\text{err}}\}$. We denote the expected reward as

$$R(p; \bar{n}) = \mathbb{E}_{Z, Z_{\text{err}}}\left[R(Z; y, Z_{\text{err}})\right].$$

In this subsection, we characterize when a higher expected reward $R(p; \bar{n})$ corresponds to a higher majority-vote accuracy, and conversely. First, we show that, conditioned on the same accuracy, the uniform error minimizes the expected reward.

**Theorem 4.5 (Uniform Error Maximizes Expected D–FUSEr Reward).** *Given an answer distribution $p \in \Delta^m$ with a fixed accuracy $p_1$, for any $\bar{n} \geq 2$, the expected reward $R(p; \bar{n})$ is maximized if and only if $e$ is uniform.*

Putting Corollary 4.4 and Theorem 4.5 together, we know that among distributions with the same accuracy, expected reward and majority-vote accuracy share the same maximizer, which is the *uniform* error distribution.

We next show that when $m = 3$, higher expected reward implies higher majority-vote accuracy, and vice versa, i.e., optimizing $r_{\text{D-FUSEr}}$ optimizes majority vote accuracy.[1]

**Theorem 4.6 (D-FUSEr Reward-Majority Accuracy Equivalence).** *Given an answer distribution $p \in \Delta^3$ with a fixed accuracy $p_1$, the majority-vote accuracy $\text{Acc}^{\text{MV}}(p; n)$ strictly increases in $R(p, \bar{n})$ for $\bar{n} \geq 2$, and vice versa.*

### 4.3. Iterative Refinement

Next, we study a setting where an LLM iteratively refines its beliefs over multiple rounds, which can be molded by a transition function $\gamma$. The model observes a sample of $n$ solutions in round $t$, and updates is solution distribution via $p^{(t+1)} = \gamma(p^{(t)}, \vec{N}^{(t)}) \in \Delta^m$, where $\vec{N}^{(t)}$ is the vote-count vector. Next, we identify conditions under which a lower error entropy leads to higher majority-vote accuracy over successive inference rounds.

---

[1] We focus on $m = 3$ as 1) error distributions are highly concentrated on the top-3 answers (Fig. 3), 2) theoretically, this restriction yields cleaner and more intuitive analysis.

**Definition 4.7** (**Monotone Updating Rule (Informal)**). An updating rule $\gamma$ is monotone if it preserves order in both error entropy and accuracy in expectation.

We defer the formal definition of monotonicity to Section B.7, and show that LLM updating is indeed monotone in figure 13. The following theorem shows that higher error entropy (i.e., higher D-FUSEr reward) leads to higher next-round majority-vote accuracy under any monotone updating rule. Since the result applies to an arbitrary round $t$, we suppress the superscript of $t$.

**Theorem 4.8** (**Higher D-FUSEr reward Increases Next-Round Majority Accuracy**). *Suppose* $m = 3$*, let* $p, p' \in \Delta^3$ *be two answer distributions such that* $p_1 = p'_1$ *and* $R(p; \bar{n}) \geq R(p'; \bar{n})$ *for an* $n \geq 2$*. If the updating rule* $\gamma$ *is monotone, then the majority accuracy of* $n \geq 1$ *i.i.d. votes drawn from* $\gamma(p, \vec{N})$ *is weakly higher than that under* $\gamma(p', \vec{N}')$ *where* $\vec{N}$ *and* $\vec{N}'$ *are the vote counts from* $p$ *and* $p'$ *respectively.*

### 4.4. Anti-Votes

As introduced in Section 3.4, anti-votes provide an inference-time mechanism for diversifying concentrated mistakes. We support this idea by showing when anti-votes are beneficial and what anti-vote distribution is desired.

For ease of analysis, we introduce a modified anti-vote rule adapted from Section 3.4. Under this rule, an anti-vote for answer $j$ does not directly penalize $j$; instead, it redistributes mass to the remaining answers via a convex combination with parameter $\alpha$. This modification is asymptotically equivalent to the anti-vote rule in Section 3.4, while enabling a simpler characterization in terms of the mixed distribution $r(\alpha) = \alpha p + (1 - \alpha)\frac{1-q}{m-1} \in \Delta^m$. We present our results in terms of $\alpha$ to align with the proofs presented in the Appendix, see Section B.9 for full details.

**Proposition 4.9** (**When Anti-Votes Help**). *Suppose* $m = 3$ *and the answer distribution is* $p = (p_1, p_2, p_3) \in \Delta^3$ *with* $p_2 > p_3$*. If* $q_1 \leq 1 - 2p_1$ *and* $q_2 \geq q_3$*, there exists an* $\alpha < 1$ *such that* $R(r(\alpha); \bar{n}) > R(r(1); \bar{n})$ *for any* $\bar{n} \geq 2$ *and* $\mathrm{Acc}^{\mathrm{MV}}(r(\alpha); n) > \mathrm{Acc}^{\mathrm{MV}}(r(1); n)$ *for any* $n \geq 3$*.*

Proposition 4.9 provides a sufficient condition under which incorporating anti-votes strictly improves performance over using votes alone. Intuitively, the requirement $q_1 \leq 1 - 2p_1$ ensures that the anti-vote predictor does not place excessive mass on the correct answer, while $q_2 \geq q_3$ means that it more reliably targets the common mistake rather than the rarer one.

In Section B.11, we characterize regimes in which anti-votes are not accurate enough to guarantee a strict improvement, and show that whenever a strict benefit is attainable, anti-votes that better predict the common mistake are always preferred.

**The Optimal Weight** $\eta$**.** We now provide a theoretical guideline of how to select the optimal $\eta$ in the original implementation described in Section 3.4. Aggregation with weight $\eta$ selects the answer maximizing $\tilde{p}_j - \eta \tilde{q}_j$, where $\tilde{p}$ and $\tilde{q}$ are the empirical distributions estimated by a finite number of samples. In the large-sample regime, correct aggregation requires that the ground-truth label $y$ be separated from all incorrect answers along the direction determined by $\eta$. This leads to the following analytical guideline:

$$\eta^\star \approx \arg\max_\eta \min_{j \neq y} \big\{ (\tilde{p}_y - \tilde{p}_j) - \eta(\tilde{q}_y - \tilde{q}_j) \big\} = \max_{j \neq y} \frac{\tilde{p}_j - \tilde{p}_y}{\tilde{q}_j - \tilde{q}_y},$$

where the maximization ranges over competitors with $q_j > q_y$. Intuitively, $\eta$ should be chosen just large enough so that the most dangerous incorrect answer—one with both high answer probability and low anti-vote probability—is down-weighted relative to $y$. This choice maximizes the worst-case margin between $y$ and all competing answers, balancing answer accuracy against anti-vote information. Formal derivations and finite-sample refinements are deferred to Section B.12.

## 5. Experiments

In this section, we present the results of our experiments. We begin by describing the experimental setup in Section 5.1, followed by a comparison of performance across models trained with the diverse failure reward in Section 5.3. We conclude by presenting our results on using anti-votes in Section 5.5. See Appendix C for full setup details and Appendix D for additional results.

### 5.1. Setup

**Datasets** We evaluate our framework on four benchmarks that include various factual and mathematical reasoning tasks: BBH (Suzgun et al., 2023), MMLU-Pro (Wang et al., 2024), GPQA (Rein et al., 2024), and RBENCH (Guo et al., 2025b).

**Models** We select a wide variety of open-source large language models to evaluate the effectiveness and reliability of our methods. We use Qwen2.5-3B-, -7B-, -14B-Instruct (Qwen et al., 2025), Gemma-2-2b-it (Google et al., 2024), and Llama3.1-8B-Instruct (Grattafiori et al., 2024).

**Baselines.** We compare our method with the following baselines: For training-free baselines, we compare with Zero-shot inference, self-reflection (Madaan et al., 2023), Zero-shot GRPO (Shao et al., 2024), and Self-Reflect GRPO (Li et al., 2025).

**Evaluations.** We present results for both in-distribution tasks and and out-of-distribution (OOD). For OOD evaluations, models are trained on a composite training de-

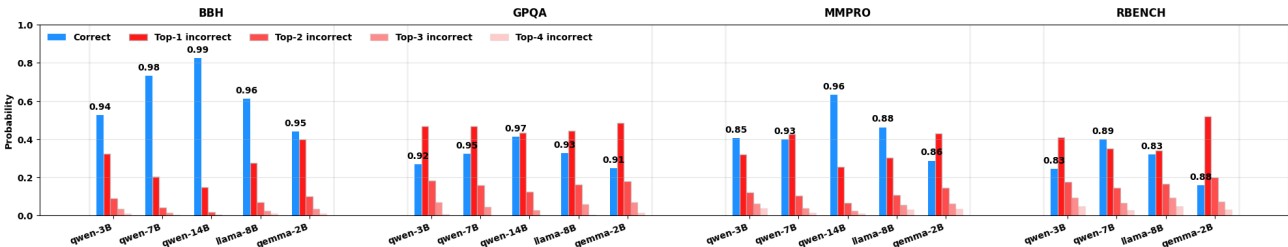

*Figure 3.* Distribution of correct and incorrect solutions from each model. The cumulative probability mass of the first three most common answers is displayed above the bar.

rived from all datasets, other than the one in question, i.e., for OOD results on BBH, models are trained on training data from MMLU-Pro, GPQA, and RBENCH. We present OOD results for all datasets. For in-distribution evaluations we maintain a disjoint training and evaluation split. Due to the small size of GPQA and RBENCH, we present in-distribution results for MMLU-Pro and BBH. All experiments are reported OOD, unless specified otherwise.

### 5.2. Distribution of Errors

Before outlining the results for our method, we first show that off-the-shelf models tend to yield *concentrated* error distributions. In Table 3 we see the distribution of solutions for both correct (blue) and the top-4 most common incorrect answers (red). In this table we make two key observations. First, we see that the vast majority of the distribution is concentrated on the first 3 most common answers, supporting our assumptions in Section 4. Second, we found that the top-1 error (i.e., the most error among all samples), consistently dominates the other top-2, -3, and -4 errors, typically taking up more than 75% of the total mass on all errors.

### 5.3. D-FUSEr: Diverse Error Reward

**Majority-Vote Accuracy** Next, we present our main results for D-FUSEr. In Table 1, we present the performance of each method when taking majority vote over 20 samples; for OOD results each method is trained using a composite training set sourced from all benchmarks *except* the listed benchmark. In these tables, we observe a common trend: models trained with our reward $r_{\text{D-FUSEr}}$ exhibit superior majority vote performance in most settings.

D-FUSEr is particularly strong in the OOD setting. The superior OOD performance of D-FUSEr is two-fold. First, D-FUSEr uses Distributional-Reflection, which provides the model a richer overview of its own prior distribution over solutions, allowing the model to better reason about the possible solution space as a whole. Second, the D-FUSEr reward teaches the model to better handle situations in which common mistakes are present by explicitly penalizing the repetition of high-frequency errors.

|  | OOD | | | | In Distribution | |
|---|---|---|---|---|---|---|
|  | BBH | GPQA | MMPRO | RBENCH | BBH | MMPRO |
| **Qwen-2.5-14B** | | | | | | |
| *— Untrained —* | | | | | | |
| Zeroshot | $87.7_{\pm0.6}$ | $44.4_{\pm1.3}$ | $68.2_{\pm0.5}$ | $58.4_{\pm0.9}$ | $87.7_{\pm0.6}$ | $68.2_{\pm0.5}$ |
| Self-reflect | $89.2_{\pm0.6}$ | $45.1_{\pm1.4}$ | $69.0_{\pm0.5}$ | $58.9_{\pm1.0}$ | $89.2_{\pm0.6}$ | $69.0_{\pm0.4}$ |
| Dis-reflect | $87.2_{\pm0.6}$ | $43.9_{\pm1.4}$ | $65.3_{\pm0.5}$ | $56.1_{\pm1.0}$ | $87.2_{\pm0.6}$ | $65.3_{\pm0.5}$ |
| *— Trained —* | | | | | | |
| GRPO | $85.2_{\pm0.6}$ | $46.7_{\pm1.3}$ | $64.7_{\pm0.5}$ | $51.7_{\pm1.1}$ | $89.9_{\pm0.5}$ | $69.5_{\pm0.4}$ |
| Reflect-GRPO | $84.3_{\pm0.6}$ | $43.3_{\pm1.3}$ | $65.3_{\pm0.5}$ | $52.5_{\pm1.0}$ | $90.1_{\pm0.5}$ | $69.2_{\pm0.5}$ |
| Dist-Reflect | $88.2_{\pm0.6}$ | $\mathbf{49.2}_{\pm1.5}$ | $70.1_{\pm0.4}$ | $60.9_{\pm1.0}$ | $\mathbf{92.3}_{\pm0.5}$ | $69.3_{\pm0.5}$ |
| D-FUSEr (ours) | $\mathbf{91.9}_{\pm0.4}$ | $48.9_{\pm1.4}$ | $\mathbf{73.5}_{\pm0.4}$ | $\mathbf{65.6}_{\pm0.9}$ | $91.4_{\pm0.5}$ | $\mathbf{75.7}_{\pm0.4}$ |
| **Llama-3.1-8B** | | | | | | |
| *— Untrained —* | | | | | | |
| Zeroshot | $69.8_{\pm0.9}$ | $36.4_{\pm1.4}$ | $55.0_{\pm0.5}$ | $40.5_{\pm0.9}$ | $69.8_{\pm0.8}$ | $55.0_{\pm0.5}$ |
| Self-Reflect | $\underline{72.3}_{\pm0.7}$ | $35.7_{\pm1.3}$ | $\mathbf{55.6}_{\pm0.5}$ | $39.3_{\pm0.9}$ | $72.3_{\pm0.8}$ | $\underline{55.6}_{\pm0.5}$ |
| Dis-Reflect | $65.7_{\pm0.9}$ | $31.6_{\pm1.4}$ | $45.6_{\pm0.5}$ | $36.7_{\pm0.9}$ | $65.7_{\pm0.8}$ | $45.6_{\pm0.5}$ |
| *— Trained —* | | | | | | |
| GRPO | $66.2_{\pm0.8}$ | $35.7_{\pm1.3}$ | $49.3_{\pm0.5}$ | $\underline{46.0}_{\pm1.0}$ | $\mathbf{77.2}_{\pm0.7}$ | $\mathbf{59.4}_{\pm0.4}$ |
| Reflect-GRPO | $64.5_{\pm0.8}$ | $\underline{38.2}_{\pm1.2}$ | $50.6_{\pm0.5}$ | $37.7_{\pm0.9}$ | $66.6_{\pm0.9}$ | $52.7_{\pm0.5}$ |
| Dist-Reflect | $71.5_{\pm0.9}$ | $33.8_{\pm1.3}$ | $48.5_{\pm0.5}$ | $37.9_{\pm0.9}$ | $74.0_{\pm0.8}$ | $50.5_{\pm0.4}$ |
| D-FUSEr (ours) | $\mathbf{73.4}_{\pm0.8}$ | $\mathbf{45.8}_{\pm1.3}$ | $52.2_{\pm0.5}$ | $\mathbf{54.9}_{\pm0.9}$ | $\underline{74.8}_{\pm0.8}$ | $53.9_{\pm0.5}$ |
| **Gemma-2-2B** | | | | | | |
| *— Untrained —* | | | | | | |
| Zeroshot | $\underline{48.0}_{\pm0.8}$ | $27.1_{\pm1.1}$ | $31.0_{\pm0.4}$ | $16.6_{\pm0.7}$ | $48.0_{\pm1.0}$ | $31.0_{\pm0.4}$ |
| Self-Reflect | $46.8_{\pm0.8}$ | $27.2_{\pm1.1}$ | $34.9_{\pm0.5}$ | $23.4_{\pm0.8}$ | $46.8_{\pm0.9}$ | $34.9_{\pm0.4}$ |
| Dis-Reflect | $44.4_{\pm1.0}$ | $26.1_{\pm1.2}$ | $31.3_{\pm0.4}$ | $25.3_{\pm0.8}$ | $44.4_{\pm0.9}$ | $31.3_{\pm0.5}$ |
| *— Trained —* | | | | | | |
| GRPO | $46.8_{\pm0.9}$ | $29.6_{\pm1.2}$ | $35.8_{\pm0.5}$ | $41.1_{\pm1.0}$ | $57.5_{\pm0.8}$ | $40.9_{\pm0.5}$ |
| Reflect-GRPO | $47.1_{\pm0.8}$ | $\underline{31.9}_{\pm1.3}$ | $34.2_{\pm0.4}$ | $37.5_{\pm0.9}$ | $59.0_{\pm0.9}$ | $40.5_{\pm0.5}$ |
| Dist-Reflect | $47.9_{\pm0.9}$ | $31.9_{\pm1.3}$ | $32.3_{\pm0.5}$ | $41.6_{\pm1.0}$ | $56.0_{\pm0.9}$ | $39.6_{\pm0.5}$ |
| D-FUSEr (ours) | $\mathbf{51.2}_{\pm0.9}$ | $\mathbf{37.7}_{\pm1.4}$ | $\mathbf{39.6}_{\pm0.5}$ | $\mathbf{42.3}_{\pm1.0}$ | $\mathbf{65.2}_{\pm0.6}$ | $\mathbf{43.0}_{\pm0.5}$ |

*Table 1.* Majority-vote accuracy (%). Best is bold; second-best is underlined (per column). Full result in Appendix D.

**Error Concentration** We now investigate a complementary question *"does $r_{\text{D-FUSEr}}$ result in less concentrated errors?"*. To measure this we define *error concentration* formally: let $\text{err}_i$ be fraction of votes for the $i^{\text{th}}$ most common error. Then error concentration is calculated as the $\ell_2^2$-norm on the top-4 most common errors, i.e.,

$$\text{error concentration} := \text{err}_1^2 + \text{err}_2^2 + \text{err}_3^2 + \text{err}_4^2$$

Note that error concentration has a maximum value of 1, meaning that all the error mass is concentrated full in $\text{err}_1$. Figure 4 confirms that this is indeed the case. We see that error concentration for D-FUSEr is typically the lowest among all training baselines. In some cases, D-FUSEr has lower error concentration than the untrained model, implying that D-FUSEr is an effective tool for reducing the solution collapse commonly associated with most post-training techniques.

| | OOD | | | | In Distribution | |
|---|---|---|---|---|---|---|
| | BBH | GPQA | MMPRO | RBENCH | BBH | MMPRO |
| **Qwen-14B** | | | | | | |
| GRPO | $84.5_{\pm0.9}$ | $45.3_{\pm2.0}$ | $64.7_{\pm0.7}$ | $49.4_{\pm1.4}$ | $85.8_{\pm0.9}$ | $65.0_{\pm0.7}$ |
| Reflect-GRPO | $83.8_{\pm0.9}$ | $41.3_{\pm1.9}$ | $64.9_{\pm0.7}$ | $51.6_{\pm1.4}$ | $86.0_{\pm0.9}$ | $65.1_{\pm0.7}$ |
| Dist-Reflect | $\underline{87.8}_{\pm0.8}$ | $\mathbf{48.4}_{\pm2.0}$ | $\underline{66.2}_{\pm0.7}$ | $\mathbf{60.5}_{\pm1.4}$ | $\mathbf{91.5}_{\pm0.7}$ | $\underline{69.2}_{\pm0.6}$ |
| D-FUSEr (ours) | $\mathbf{89.4}_{\pm0.8}$ | $\underline{47.5}_{\pm2.0}$ | $\mathbf{69.8}_{\pm0.6}$ | $\underline{59.9}_{\pm1.4}$ | $\underline{89.5}_{\pm0.8}$ | $\mathbf{71.7}_{\pm0.6}$ |
| **Llama-8B** | | | | | | |
| GRPO | $\underline{65.2}_{\pm1.2}$ | $32.5_{\pm1.8}$ | $\underline{47.5}_{\pm0.7}$ | $\mathbf{45.3}_{\pm1.4}$ | $\mathbf{70.8}_{\pm1.2}$ | $\mathbf{52.9}_{\pm0.7}$ |
| Reflect-GRPO | $59.2_{\pm1.2}$ | $36.0_{\pm1.9}$ | $46.3_{\pm0.7}$ | $37.0_{\pm1.3}$ | $66.1_{\pm1.2}$ | $50.0_{\pm0.7}$ |
| Dist-Reflect | $\mathbf{69.8}_{\pm1.2}$ | $33.7_{\pm1.9}$ | $37.0_{\pm0.7}$ | $37.9_{\pm1.3}$ | $\underline{68.1}_{\pm1.2}$ | $50.4_{\pm0.7}$ |
| D-FUSEr (ours) | $62.2_{\pm1.2}$ | $\mathbf{39.4}_{\pm1.9}$ | $\mathbf{50.6}_{\pm0.7}$ | $\underline{42.7}_{\pm1.4}$ | $66.5_{\pm1.2}$ | $\underline{51.9}_{\pm0.7}$ |
| **Gemma-2B** | | | | | | |
| GRPO | $45.4_{\pm1.3}$ | $27.9_{\pm1.8}$ | $\underline{35.5}_{\pm0.7}$ | $40.6_{\pm1.4}$ | $55.4_{\pm1.3}$ | $37.1_{\pm0.7}$ |
| Reflect-GRPO | $46.3_{\pm1.3}$ | $31.5_{\pm1.8}$ | $33.7_{\pm0.7}$ | $36.3_{\pm1.3}$ | $54.5_{\pm1.3}$ | $37.7_{\pm0.7}$ |
| Dist-Reflect | $\underline{46.8}_{\pm1.3}$ | $31.9_{\pm1.8}$ | $31.8_{\pm0.6}$ | $\mathbf{41.5}_{\pm1.4}$ | $55.2_{\pm1.3}$ | $38.5_{\pm0.7}$ |
| D-FUSEr (ours) | $\mathbf{47.0}_{\pm1.3}$ | $\mathbf{34.9}_{\pm1.9}$ | $\mathbf{37.6}_{\pm0.7}$ | $40.5_{\pm1.4}$ | $\mathbf{61.5}_{\pm1.2}$ | $\mathbf{41.4}_{\pm0.7}$ |

*Table 2.* Average accuracy (%) at the final round. Best is bold and second-best is underlined.

**Average Accuracy.** Table 2 reports the average accuracy of each method. We first observe that although D-FUSEr is designed to optimize majority-vote accuracy, it also improves average accuracy, outperforming accuracy-only baselines in some settings. That said, D-FUSEr's advantage is less pronounced for average accuracy than for majority vote. This observation further supports our hypothesis that optimizing for test-time scaling with majority vote requires explicitly accounting for the distribution of errors, rather than correctness alone: even when its average accuracy gains are more modest, D-FUSEr consistently achieves stronger improvements in majority-vote accuracy.

Second, Dist-Reflection trained solely with a correctness objective outperforms both GRPO and Reflect GRPO. In Dist-Reflection, the model conditions on an approximation of its own solution distribution when generating new responses. We hypothesize that the success of of Dist-Reflection (even without $r_{\text{D-FUSEr}}$ is due to the fact that the model is able to observe its own prior during training; allowing for greater exploration. This exploration is further compounded when also using $r_{\text{D-FUSEr}}$. We attribute the increased average accuracy of D-FUSEr to these two complementary incentives for exploration: exposure to diverse solutions through Dist-Reflection and an explicit reward signal that encourages diverse failures.

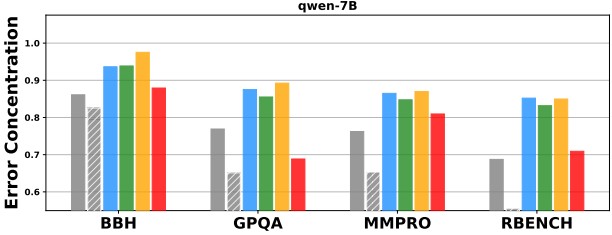

*Figure 4.* Error Concentration for Qwen-2.5-3B

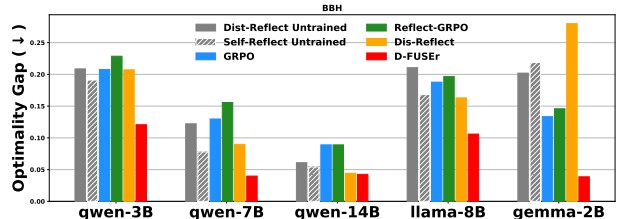

*Figure 5.* Optimality gap, measured in terms of difference in majority vote accuracy.

### 5.4. Optimality Gap

Lastly, we study the optimality gap of each method with respect to the theoretical optimum policy under $r_{\text{D-FUSEr}}$, which is derived from Theorem 4.5. While the underlying policy is complex (an LLM), optimality is defined over the solution distribution. In particular, the theorem implies that the optimal solution distribution assigns uniform mass over incorrect answers. This makes the theoretical optimum easy to compute: for each method, we keep all correct predictions fixed and redistribute the probability mass of errors uniformly over the top-4 incorrect answers. The optimality gap is then defined as the difference between the majority-vote accuracy of this optimized distribution and the observed majority-vote accuracy.

Across all datasets and methods, we observe a non-trivial optimality gap, indicating that increased error uniformity would improve majority-vote performance for every policy. Notably, D-FUSEr consistently achieves the smallest optimality gap, implying that its induced error distribution is closest to the theoretical optimum. This shows that D-FUSEr not only improves accuracy, but does so by shaping errors in a manner aligned with the optimal policy characterized by our theory.

### 5.5. Results: Anti-votes

We illustrate the effect of anti-votes on the RBENCH dataset in Figure 6. Results for all datasets are provided in Appendix E. In each plot, Anti-vote (Fixed) denotes the best performance achieved using a fixed anti-vote strength ($\eta$), while Anti-vote (Dynamic) corresponds to the performance obtained by applying our theoretically derived anti-vote formula, which is question-dependent. For each question, 20 votes and 10 anti-votes were used. As shown in the figure, anti-votes consistently improve majority-vote accuracy across datasets and model sizes, demonstrating substantial potential for mitigating error concentration. We also note that these results are obtained using **off-the-shelf models**, indicating that majority-vote aggregation can benefit substantially from anti-votes without additional training.

While the theoretical formula may not be easily operationalized in practice, these results indicate that majority-vote

*Figure 6.* **Experimental Results: Anti-votes.** Majority-vote accuracy with and without anti-votes across all models (round 0, without distributional reflection). Anti-vote (Fixed) reports the best performance achieved using a fixed anti-vote strength $\eta$, and Anti-vote (Dynamic) applies our theoretical formula for optimal anti-vote strength. Anti-votes consistently improve majority-vote accuracy over the baseline, highlighting the potential of anti-votes for mitigating error concentration. Results for all datasets are provided in Appendix E.

aggregation could benefit significantly from anti-votes given a sufficiently accurate predictor. We leave the problem of training such a predictor for future work.

## 6. Related Works

**Inference-time Scaling.** Inference-time scaling refers to a class of techniques that improve model performance by allocating additional computation at inference rather than at training time. One of the earliest and most influential examples of inference-time scaling is chain-of-thought (CoT) prompting (Wei et al., 2022; Kojima et al., 2022), which encourages models to reason step by step. CoT prompting has since become a foundation for many state-of-the-art reasoning models (Guo et al., 2025a).

Subsequent work on inference-time scaling can be broadly categorized into two groups. The first focuses on *sampling and aggregation*; models generate multiple outputs and combine them using majority vote or self-consistency (Wang et al., 2023; Yao et al., 2023; Brown et al., 2024; Yao et al., 2023). The second emphasizes *iterative refinement*; models improve an answer through multiple rounds of self-feedback, critique, or structured search (Madaan et al., 2023; Zhou et al., 2024; Yao et al., 2022; Du et al., 2023). However, both approaches largely treat the model's error distribution as fixed and do not explicitly address correlated failures. Our work complements inference-time scaling methods by focusing on how errors are distributed across samples.

**Improving LLM Reasoning Diversity.** Another line of related work focuses on encouraging diversity in LLM responses. This literature can be broadly categorized into several groups. The first includes prompting-based methods that promote response diversity, such as prompt perturbation or paraphrasing (Naik et al., 2023; Wang et al., 2025b), multilingual prompting (Wang et al., 2025a), or prompting models to reason using distinct methodologies or target different final answers (Liu et al., 2025a; Liang et al., 2024; Hegazy, 2024). The second category explores branching and search over intermediate reasoning states, as in Tree-of-Thoughts (Yao et al., 2023) or Monte Carlo Tree Search–

based approaches (Zheng et al., 2025). A third line of work focuses on training models to improve such branching, often using process-level reward models that score intermediate reasoning steps (Chia et al., 2024; Khalifa et al., 2026; Li & Li, 2025; Yang et al., 2025). Finally, model merging (Yang et al., 2024) and token-level mixing of multiple models during generation (Sanyal et al., 2025) aim to increase diversity by combining multiple models at the token or weights level during inference.

Despite the breadth of work aimed at improving reasoning diversity, relatively few studies examine how diversity interacts with downstream aggregation mechanisms such as majority vote. In contrast, we explicitly identify and analyze this connection, showing that the distribution of errors plays a central role in aggregation performance. Our approach directly addresses this gap by shaping the error distribution through a novel reward scheme that encourages more uniform error distribution. Lastly, we note that our work is also related to the uncertainty quantification literature (e.g., Kuhn et al. (2023)), where researchers studied measuring uncertainty over a *distribution* of responses.

## 7. Conclusion

We introduced **D-FUSEr**, a unified framework for improving inference-time scaling by explicitly controlling how errors are distributed across samples. Starting from *distributional reflection*, we provided two complementary mechanisms for reducing error concentration in inference-time scaling: (a) diverse failure reward, a theoretically grounded training objective which encourages calibrated and diverse errors while preserving single-sample accuracy, and (b) anti-votes, an inference-time mechanism in which the model explicitly predicts likely incorrect answers alongside its solution. Empirically, across three model families of varying sizes and four benchmarks, we showed that both approaches substantially improve majority vote and iterative refinement performance without degrading single-sample accuracy. Together, these results demonstrate that shaping error distributions is a practical and theoretically grounded way to improve both aggregation and refinement at infer-

ence time. We hope our perspective motivates future work on aggregation-aware training and self-reflection methods that explicitly account for the error distributions.

**Limitations.** We note that our analysis relies on discrete answer spaces with well-defined answer extraction, making generalization to open-ended settings unclear. Moreover, distributional reflection introduces extra inference and training cost. Anti-votes require a sufficiently accurate predictor for full potential, which we leave for future work. For open-ended questions, clustering algorithms or LLM-based categorization could enable our framework, but this deserves separate investigation. We further discuss the limitations of our theoretical framework in Section B.1.

## Impact Statement

This work introduces D-FUSEr, a training pipeline that shapes the distribution of LLM errors to improve the efficacy of test time scaling (majority vote). While this can increase efficacy by reducing the impact of shared errors, it does not completely remove errors. De-corelating errors can also make model responses more noisy, models should be tested for such pathologies before deployment. Beyond shaping errors, it is important to note that D-FUSEr does not expand model capabilities or introduce new behavioral modes beyond those of the base model, and will have similar impacts to other reasoning based post-training methods.

## Conflict of Interest Disclosure

We declare that we have no conflicts of interest relevant to the content of this article.

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

## A. Direct majority-vote accuracy optimization is insufficient

A natural question to ask is, why use a proxy reward like DFUSER instead of optimizing majority vote directly. We find there is both an empirical and conceptual issue with this direct optimization. In Table 9 we show that directly optimizing majority vote accuracy leads to worse performance. We next outline "why" this is a case with an illustrative example.

**Why optimizing majority vote directly fails.** Consider computing the contribution of a single new answer $S_{\text{new}}$ given other answers $S_1, \ldots, S_M$ (with $M$ even for simplicity). The new answer can only change the majority vote outcome in two settings:

1. Exactly $M/2$ of the existing answers are correct and $M/2$ are incorrect. Here, $S_{\text{new}}$ is the tiebreaker, if it is correct, the majority vote is correct, and vice versa.

2. Exactly $M/2 - 1$ answers fall on one side. Then $S_{\text{new}}$ can either create or break a tie.

Crucially, both settings are unlikely events, they require the existing pool to be near-perfectly split. Moreover, even if we encounter the cases above, the reward reduces to rewarding individual correctness, since $S_{\text{new}}$ single-handedly dictates the outcome. The result is that directly optimizing majority vote accuracy collapses to a sparse version of an average accuracy reward.

Alternatively, we could reward based on the number of correct answers in the majority. This would get rid of the sparsity issue, but we again recover something very similar to average accuracy.

**Why use D-FUSEr.** This is precisely what motivated D-FUSEr: rather than trying to optimize the majority vote objective directly, we target the error distribution. This provides a denser training signal that is theoretically guaranteed to improve majority vote accuracy (Theorem 4.6). Designing a more sophisticated technique to make direct majority vote optimization viable is an interesting direction.

Below we include a comparison of directly optimizing majority vote for Qwen-2.5-14B-Instruct: As predicted, Direct-Maj underperforms Dist-Reflect, confirming the credit assignment issue.

| | OOD | | | | In Distribution | |
|---|---|---|---|---|---|---|
| | BBH | GPQA | MMPRO | RBENCH | BBH | MMPRO |
| — Untrained — | | | | | | |
| Zeroshot | 87.7 | 44.4 | 68.2 | 58.4 | 87.7 | 68.2 |
| Self-Reflect | 89.2 | 45.1 | 69.0 | 58.9 | 89.2 | 69.0 |
| Dist-Reflect | 87.2 | 43.9 | 65.3 | 56.1 | 87.2 | 65.3 |
| — Trained — | | | | | | |
| GRPO | 85.2 | 46.7 | 64.7 | 51.7 | 89.9 | 69.5 |
| Reflect-GRPO | 84.3 | 43.3 | 65.3 | 52.5 | 90.1 | 69.2 |
| Dist-Reflect + Direct-Maj (new) | 87.1 | 44.9 | 66.2 | 57.8 | 87.0 | 67.2 |
| Dist-Reflect | 88.2 | **49.2** | 70.1 | 60.9 | **92.3** | 69.3 |
| D-FUSEr (ours) | **91.9** | 48.9 | **73.5** | **65.6** | 91.4 | **75.7** |

*Table 3.* Majority-vote accuracy (%) for Qwen-2.5-14B. Best is bold.

## B. Theoretical Results and Deferred Proofs

### B.1. Assumptions

We first provide additional context for the assumptions used in our theoretical results.

**Assumption of three alternative answers** Our analysis fixes $m = 3$ for clarity, but the conclusion of Theorem 4.6 extends approximately to larger answer spaces. Suppose the mass on mistakes outside the top two incorrect answers is bounded, $p_4 + \cdots + p_m \leq \varepsilon$ for some small $\varepsilon$. Then, in the worst case, the entire $\varepsilon$ mass concentrates on a single incorrect answer, maximizing its chance of outvoting the correct answer, and the resulting loss in majority accuracy is bounded by something

proportionally small to $\varepsilon$. Since the top three answers empirically carry nearly all the probability mass (Figure 15), one can expect that $\varepsilon$ is small in practice.

**Monotonicity assumption** We do not expect LLMs to satisfy the monotone update rule exactly, but both our experiments and prior work indicate it holds approximately at the answer level. Figure 13 shows that error concentration increases across refinement rounds, consistent with monotone updating. Estornell & Liu (2024) similarly find that in multi-agent debate, majority opinions at one round are likely to be reinforced at the next round, and Oh et al. (2025) describe such systems as an "echo chamber" for majority opinions.

**Fixed single-sample accuracy assumption** Our theory holds single-sample accuracy $p_1$ fixed in order to isolate the effect of the error distribution on majority-vote accuracy. This is a modeling choice rather than a property of our method: in practice the training objective combines the correctness reward with the diverse failure reward, so the model is incentivized to preserve accuracy while diversifying its errors. The theory thus characterizes why error shaping helps when accuracy is held constant, and the empirical results confirm that the composite objective improves majority-vote accuracy without degrading single-sample accuracy.

## B.2. Proof of Proposition 4.3

*Proof.* Recall that we use $\vec{N}$ to denote vote counts. Conditioning on the number of correct answers $N_1 = n_1$, the error-count vector $N_{-1} := (N_2, \ldots, N_m)$ follows a multinomial distribution Multinomial$(n - n_1; e)$. Let $\max(N_{-1}) = \max_{j \geq 2} N_j$ be the error plurality. Then, the majority vote of the $n$ answers agrees with the ground truth if and only if $\max(N_{-1}) < n_1$ (assuming worst-case tie-breaking). The following lemma shows that, for any fixed number of correct answers $n_1$, this event becomes more likely when the error distribution has a lower entropy.

**Lemma B.1.** *If $e' \succeq e$, then for any $n \geq 1$ and $0 \leq n_1 \leq n$,*

$$\Pr(\max(N_{-1}) < n_1) \ \geq \ \Pr\big(\max(N'_{-1}) < n_1\big),$$

*where $N_{-1} \sim$ Multinomial$(n - n_1; e)$ and $N'_{-1} \sim$ Multinomial$(n - n_1; e')$.*

Lemma B.1 suggests that majorization is a sufficient condition for an error distribution being less desirable than another, as it has a higher probability of reaching an incorrect majority answer. The following result formally states this.

The proposition then follows from Lemma B.1 by taking the expectation over $N_1$.

By Lemma B.1, since $e' \succeq e$, for every $n \geq 1$ and every $0 \leq n_1 \leq n$,

$$\Pr\big(\max(N_{-1}) < n_1\big) \ \geq \ \Pr\big(\max(N'_{-1}) < n_1\big).$$

Therefore, for any $n_1 \in \{0, \ldots, n\}$,

$$\Pr_e(\text{Plurality} = 1 \mid N_1 = n_1) \ \geq \ \Pr_{e'}(\text{Plurality} = 1 \mid N_1 = n_1).$$

Finally, taking expectation over $N_1$ and using the law of total probability completes the proof. $\square$

*Proof of Lemma B.1.* First, Schur (1923) established the condition which suggests that, for any Schur-concave function $\phi$, $v' \succeq v$ implies that $\phi(v') \leq \phi(v)$. Therefore, the key to the proof is showing that $\phi(e) := \Pr(\max(N_{-1}(e)) < n_1)$ is Schur-concave in $e$ with $N_{-1} \sim$ Multinomial$(n - n_1; e)$. We can explicitly write down the analytical expression of $\phi(e)$ given the multinomial variable. Let $\bar{S} = \{n_{-1} = (n_2, \ldots, n_m) \in \{0, \ldots, n_1 - 1\}^{m-1} \mid \sum_{i=2}^{m} n_i = n - n_1\}$. To avoid ambiguity, we index the entries of vector $e$ with the answer ID, with $e_i = p_i/(1 - p_1)$ for $i = 2, \ldots, m$.

$$\phi(e) = \Pr\big(N_2 < n_1, \ldots, N_m < n_1\big) \tag{1}$$

$$= (n - n_1)! \sum_{n_{-1} \in \bar{S}} \prod_{k=2}^{m} \frac{e_k^{n_k}}{n_k!}. \tag{2}$$

Let $I_k = \{s = (s_1, \ldots, s_n) \in \mathbb{Z}^n \mid \sum_{i=1}^{n} s_i = k \text{ and } s_i \geq 0 \ \forall i\}$. A subset $S \subset I_k$ is said to be *Schur-concave index set* if $s \in S$, $s' \in I_k$ and $s \succeq s'$, then $s' \in S$. Marshall et al. (1979) prove the following result, which directly applies to the Schur-concavity of the multinomial distribution.

**Lemma B.2** ((Marshall et al., 1979), Example 3.J.2.b). *Let $k, n \geq 1$ and let $S \subset I_k$ be a Schur-concave index set. Then, the polynomial $H_S(p_1, \ldots, p_n) = \sum_{s \in S} \Pi_i^n \frac{p_i^{s_i}}{s_i!}$ is Schur-concave on $[0, \infty)^n$.*

It is straightforward to see that $\bar{S}$ in Equation (1) is a Schur-concave index set. Therefore, Theorem B.2 implies $\phi(e)$ being Schur-concave in $e$, which completes the proof. $\qquad\square$

### B.3. Proof of Proposition 4.5

*Proof.* We first rewrite the expected reward considering the randomness of the new sample $Z$. Because $Z$ and $Z_{\text{err}}$ are independent,

$$R(p; \bar{n}) = p_1 - \lambda \Pr(Z = Z_{\text{err}}) = p_1 - \lambda(1 - p_1)\mathbb{E}[e_{Z_{\text{err}}}].$$

Because $p_1$ is fixed, we only have to show that $\mathbb{E}[e_{Z_{\text{err}}}]$ is minimized iff $e$ is uniform.

Let $K$ be the number of incorrect answers among the $n$ reference samples. Then $K \sim \text{Bin}(\bar{n}, 1 - p_1)$, and conditional on $K = k$, the incorrect counts $(N_2, \ldots, N_m)$ satisfy

$$(N_2, \ldots, N_m) \mid (K = k) \sim \text{Multinomial}(k; e).$$

Let $J_k(e) \in \{2, \ldots, m\}$ denote the (uniform-tie-broken) index of the most frequent incorrect answer among these $k$ samples. Because $Z$ is independent of the reference samples,

$$\Pr(Z = Z_{\text{maj}}) = \mathbb{E}[p_{J_K(e)}] = (1 - p_1)\sum_{k=0}^{n} \Pr(K = k)\, F_k(e),$$

where $F_k(e) := \mathbb{E}[e_{J_k(e)}]$. Because $\Pr(K = k)$ is independent of $e$, it suffices to show that for each $k$, $F_k(e)$ is minimized if and only if $e$ is uniform.

We prove this using the idea of induction. First, if $k = 0$, $J_0(e)$ is uniformly selected and $F_0(e)$ is independent of $e$. Fix $k \geq 1$ and choose two indices $a, b \in \{2, \ldots, m\}$. Define $e^{\text{avg}}$ by replacing $(e_a, e_b)$ with their average:

$$e_a^{\text{avg}} = e_b^{\text{avg}} = \frac{e_a + e_b}{2}, \qquad e_j^{\text{avg}} = e_j \;\; (j \neq a, b).$$

The following lemma shows that $e^{\text{avg}}$ is always preferred.

**Lemma B.3.** *$F_k(e^{\text{avg}}) \leq F_k(e)$, and the inequality is strict for $k \geq 2$ unless $e_a = e_b$.*

Starting from any $e$, repeatedly pick two coordinates that are unequal and replace them by their average. By Lemma B.3, this operation strictly decreases $F_k(e)$ as long as $k \geq 2$, which happens with a positive probability because $\bar{n} \geq 2$. Iteratively applies this operation leads to the uniform distribution and along the way $F_k$ strictly decreases. This completes the proof. $\qquad\square$

*Proof of Lemma B.3.* Suppose that among the $k$ incorrect samples, $S = N_a + N_b$ of them fall into $\{a, b\}$. Then, conditioned on $S = s$, we have

$$N_a \mid S = s \sim \text{Bin}\left(s, u\right), \qquad u := \frac{e_a}{e_a + e_b},$$

while under $e^{\text{avg}}$ the corresponding parameter is $u = \frac{1}{2}$. Crucially, the distribution of $S$ is identical under $e$ and $e^{\text{avg}}$. Therefore, whenever the majority mistake is neither $a$ nor $b$, $J_k(e) = J_k(e^{\text{avg}})$ and the target function is the same under the two error distributions.

Let $T = \max_{j \geq 2, j \neq a, b} N_j$ be the maximum number of votes for the incorrect answers that are not $a, b$. Conditioned on $T = t \in \{0, \ldots, k - s\}$, the only randomness comes from $N_a$, i.e. the binomial distribution between the votes in $\{a, b\}$. Let $t' = \max(t, \lceil \frac{s}{2} \rceil)$, it is sufficient to show that

$$f(u) := u \Pr_u(N_a \geq t') + (1 - u) \Pr_u(N_a \leq s - t'), \tag{3}$$

is minimized when $u = \frac{1}{2}$ for any $s, t$, and is strictly minimized for some $s, t$. Here, $f(u) = F_k(e) \mid s, t$ where the condition $s$ is on the number of samples in $\{a, b\}$ and $t$ is the number of samples for the majority mistake that is not $a, b$. We prove the property of $f$ in two steps.

**$f(u)$ is symmetric.**

Define

$$A(u) := \Pr_u(N_a \geq t').$$

By symmetry of the binomial distribution,

$$\Pr_u(N_a \leq s - t') = \Pr_{1-u}(N_a \geq t') = A(1 - u),$$

and therefore

$$f(u) = uA(u) + (1 - u)A(1 - u).$$

This immediately implies the symmetry $f(u) = f(1 - u)$, meaning that $u = \frac{1}{2}$ is the only possible interior minimizer.

**$f(u)$ is strictly increasing on $u \in (\frac{1}{2}, 1)$.**

Suppose $1 \leq t' \leq s$, which happens with a positive probability because $k \geq 2$. Then, it is easy to see that the binomial tail $A(u)$ is strictly increasing in $u$ on $(0, 1)$:

$$A'(u) = \frac{d}{du} \Pr_u(N_a \geq t') = s\binom{s-1}{t'-1} u^{t'-1}(1-u)^{s-t'} > 0,$$

for all $u \in (0, 1)$.

Next, differentiating $f(u) = uA(u) + (1 - u)A(1 - u)$ gives

$$\begin{aligned}
f'(u) &= A(u) + uA'(u) - A(1-u) - (1-u)A'(1-u) \\
&= \big(A(u) - A(1-u)\big) + \big(uA'(u) - (1-u)A'(1-u)\big).
\end{aligned}$$

Since $A$ is strictly increasing and $u > 1 - u$ for $u > \frac{1}{2}$, the first term $(A(u) - A(1-u)) > 0$ for any $u > \frac{1}{2}$. It remains to show that the second term is non-negative. Let $C := s\binom{s-1}{t'-1} > 0$. Then

$$\begin{aligned}
uA'(u) &= C\, u^{t'}(1-u)^{s-t'}, \\
(1-u)A'(1-u) &= C\,(1-u)^{t'} u^{s-t'}.
\end{aligned}$$

Thus $uA'(u) \geq (1-u)A'(1-u)$ is euuivalent to $u^{t'}(1-u)^{s-t'} \geq (1-u)^{t'} u^{s-t'}$. Dividing by the positive term $u^{s-t'}(1-u)^{s-t'}$ yields

$$u^{2t'-s} \geq (1-u)^{2t'-s}.$$

Since $u > \frac{1}{2}$ and $2t' - s \geq 0$ by definition, the inequality holds.

We have shown that $f(u)$ is strictly increasing on $u \in [\frac{1}{2}, 1]$. By symmetry, $f(u)$ is strictly decreasing $u \in [0, \frac{1}{2}]$. Therefore, $f(u)$ is strictly minimized at $u = \frac{1}{2}$, and the minimizer is unique. $\qquad\square$

### B.4. Proof of Theorem 4.6

We first present the following lemma.

**Lemma B.4** (Strict monotonicity of MV accuracy for $m = 3$)**.** *Let $p = (\rho, (1-\rho)u, (1-\rho)(1-u)) \in \Delta^3$ with $\rho \in [0, 1]$ and $u \in [1/2, 1]$. Let $\mathrm{Acc}^{\mathrm{MV}}(p(\rho, u); n)$ denote the majority-vote accuracy of $n$ i.i.d. draws from $p$ with uniform tie-breaking. Then:*

1. *$\mathrm{Acc}^{\mathrm{MV}}(p(\rho, u); n)$ is strictly increasing in $\rho$ for fixed $u$ if $n \geq 1$.*

2. *$\mathrm{Acc}^{\mathrm{MV}}(p(\rho, u); n)$ is decreasing in $u$ for fixed $\rho$ if $n \geq 1$, and strictly decreasing if $n \geq 3$.*

*Proof of Lemma B.4.* Let $p = (\rho, (1 - \rho)u, (1 - \rho)(1 - u))$ and draw $n$ i.i.d. samples. Let $k$ denote the *total number of mistakes*, i.e.,

$$k := n - N_1,$$

so $k \sim \text{Bin}(n, 1 - \rho)$ and $N_1 = n - k$.

Condition on $k$. Then the $k$ mistakes are i.i.d. over $\{2, 3\}$ with probabilities $(u, 1 - u)$, hence

$$N_2 \mid k \sim \text{Bin}(k, u), \qquad N_3 = k - N_2.$$

Define the conditional majority-vote accuracy

$$g_k(\rho, u) := \Pr(\text{MV outputs label } 1 \mid k).$$

Then

$$\text{Acc}^{\text{MV}}(p(\rho, u); n) = \sum_{k=0}^{n} \Pr\big(\text{Bin}(n, 1 - \rho) = k\big) \, g_k(\rho, u). \tag{4}$$

**Part (1): strictly increasing in $\rho$ (for fixed $u$, $n \geq 1$).** Fix $u$. For each $k$, label 1 has $n - k$ votes and the others have $k$ votes in total. If we decrease $k$ by 1 (equivalently increase $N_1$ by 1), the majority advantage of label 1 weakly improves for every realization of $(N_2, N_3)$, hence $g_k(\rho, u)$ is (weakly) decreasing in $k$. Moreover, $g_0(\rho, u) = 1$ and $g_n(\rho, u) = 0$, so the sequence $\{g_k(\rho, u)\}_{k=0}^{n}$ is not constant.

Now as $\rho$ increases, $1 - \rho$ decreases, so the binomial $k \sim \text{Bin}(n, 1 - \rho)$ shifts stochastically to smaller values. Since $g_k(\rho, u)$ is decreasing in $k$ and not constant, the expectation in equation 4 is strictly increasing in $\rho$.

**Part (2): nonincreasing in $u$ (for fixed $\rho$, $n \geq 1$).** Fix $\rho$ and $n \geq 1$. Given the number of total mistakes $k$, MV outputs label 1 iff label 1's count $n - k$ is at least the maximum of $N_2$ and $k - N_2$ (with uniform tie-breaking on ties). Equivalently, label 1 wins iff

$$N_2 \in [k - (n - k), \ n - k] = [2k - n, \ n - k].$$

Let

$$t := n - k, \qquad a := \max(0, 2k - n), \qquad b := \min(k, t).$$

Then $g_k(\rho, u)$ can be written as a weighted central-binomial mass

$$g_k(\rho, u) = \sum_{x=a}^{b} w_x \binom{k}{x} u^x (1 - u)^{k - x}, \tag{5}$$

where $w_x \in (0, 1]$ are the (uniform) tie-breaking weights, and $w_x = w_{k-x}$ (swapping labels 2 and 3 preserves the rule and tie-breaking). By the symmetry argument in Lemma B.3, this weighted central binomial mass is nonincreasing in $u$ on $[1/2, 1]$. Taking expectation over $k$ in equation 4 shows that $\text{Acc}^{\text{MV}}(p(\rho, u); n)$ is nonincreasing in $u$ for all $n \geq 1$.

**Part (3): strictly decreasing in $u$ (for fixed $\rho$, $n \geq 3$).** If $n \geq 3$, then with positive probability $k \in \{1, \ldots, n - 1\}$ and the winning region $\{a, \ldots, b\}$ is nontrivial. Lemma B.3 then implies that $g_k(\rho, u)$ is *strictly* decreasing in $u$ for such $k$ on $(1/2, 1]$. Therefore $\text{Acc}^{\text{MV}}(p(\rho, u); n)$ is strictly decreasing in $u$.

$\square$

*Proof of Theorem 4.6.* Without loss of generality, suppose $p_2 \geq p_3$, and let $p(u) = (p_1, (1 - p_1)u, (1 - p_1)(1 - u))$ where $u \in [\frac{1}{2}, 1]$. Let $N = (N_1, N_2, N_3) \sim \text{Multinomial}(n; p(u))$ be vote counts from $n$ i.i.d./ draws. To establish the claim, it suffices to show that if $n \geq 3$, as $u$ increases on $[\frac{1}{2}, 1]$,

1. The accuracy of the majority vote $\text{Acc}^{\text{MV}}(p(u); n)$ is strictly decreasing; and

2. the expected reward $R(p(u); \bar{n})$ is strictly decreasing.

The first point follows from Lemma B.4.

For the second point, let $S \in [\bar{n}]$ denote the number of incorrect reference samples and condition on $S = s \geq 1$. Write $N_2 \mid (S = s) \sim \text{Bin}(s, u)$ and $t := s/2$. A direct calculation yields

$$R(p(u); \bar{n}) = p_1 - \lambda(1 - p_1) D(u),$$

where, conditioning on $S = s$,

$$D(u) \mid (S = s) := u \Pr(X > t) + (1 - u) \Pr(X < t) + \tfrac{1}{2} \Pr(X = t).$$

Let $t' := \lfloor t \rfloor + 1 = \lfloor s/2 \rfloor + 1$. Then $\Pr(X > t) = \Pr(X \geq t')$ and $\Pr(X < t) = \Pr(X \leq s - t')$. Hence

$$D(u) \mid (S = s) = u \Pr_u(X \geq t') + (1 - u) \Pr_u(X \leq s - t') + \tfrac{1}{2} \Pr_u(X = t). \qquad (6)$$

Comparing equation 6 with equation 3 in the proof of Lemma B.3, we are exactly in the same "symmetric binomial tail + mixture weight" form (with parameter $u$ and the above $t'$). Therefore, for every fixed $s \geq 1$ and $t' \geq 1$ (meaning that $\bar{n} \geq 2$) the quantity $D(u) \mid (S = s)$ is *strictly increasing* in $u$ on $(\tfrac{1}{2}, 1]$.

Since $\Pr(S \geq 1 \text{ and } \bar{n} - S \geq 1) > 0$ when $\bar{n} \geq 2$, taking expectation over $S$ preserves strictness: $D(u) = \mathbb{E}[D(u) \mid S]$ is strictly increasing in $u$ on $(\tfrac{1}{2}, 1]$. Consequently,

$$R(p(u); \bar{n}) = p_1 - \lambda(1 - p_1)D(u)$$

is strictly *decreasing* in $u$ on $(\tfrac{1}{2}, 1]$.

This implies the monotonicity in the expected reward and the majority-vote accuracy, which completes the proof. □

## B.5. Proof of Theorem 4.8

*Proof.* Let $(p_1^+, (1 - p_1^+)e^+) := \gamma(p, \vec{N})$ and $(p_1'^+, (1 - p_1'^+)e'^+) := \gamma(p', \vec{N}')$. By the following lemma, we can couple $(\vec{N}, \vec{N}')$ so that $N_1' = N_1$ and $\max(N_2', N_3') \geq \max(N_2, N_3)$ almost surely. Under this coupling, $\epsilon' = (N_2', N_3')/(n - N_1') \succeq \epsilon = (N_2, N_3)/(n - N_1)$ almost surely. Then, the monotonicity of $\gamma$ implies $p_1^+ \geq p_1'^+$ $e'^+ \succeq e^+$ and $p_1^+ \geq p_1'^+$.

**Lemma B.5** (Multinomial coupling for $m = 3$). *Let $p = (p_1, (1 - p_1)q, (1 - p_1)(1 - q))$ and $p' = (p_1, (1 - p_1)q', (1 - p_1)(1 - q'))$ with $q, q' \in [1/2, 1]$ and $q' \geq q$. Let $\vec{N} \sim \text{Mult}(n; p)$ and $\vec{N}' \sim \text{Mult}(n; p')$. Then there exists a coupling of $(\vec{N}, \vec{N}')$ such that*

$$N_1' = N_1 \quad and \quad \max(N_2', N_3') \geq \max(N_2, N_3) \qquad a.s.$$

It remains to show that the next-round majority-vote accuracy is higher under $p^+$ than that under $p'^+$. For $m = 3$, $e'^+ = (u'^+, 1 - u'^+) \succeq e^+ = (u^+, 1 - u^+)$ means that the updated incorrect-label concentration parameter satisfies $u'^+ \geq u^+$. By Lemma B.4, the majority-vote accuracy is nondecreasing in $p_1$ and nonincreasing in $u$.

Therefore,

$$\text{Acc}_{\text{MV}}(\gamma(p, \vec{N}); n) \geq \text{Acc}_{\text{MV}}(\gamma(p', \vec{N}'); n) \quad \text{almost surely under the coupling.}$$

Taking expectations over the coupling completes the proof. □

*Proof of Lemma B.5.* Since $p_1 = p_1'$, draw $N_1 \sim \text{Bin}(n, p_1)$ and set $N_1' = N_1$. Let $k = n - N_1$. Conditional on $N_1$, the remaining counts satisfy $N_2 \sim \text{Bin}(k, q)$ and $N_2' \sim \text{Bin}(k, q')$, with $N_3 = k - N_2$ and $N_3' = k - N_2'$. Define $M = \max(N_2, N_3) = \max(N_2, m - N_2)$ and $M' = \max(N_2', N_3') = \max(N_2', m - N_2')$.

For any $t \geq \lceil m/2 \rceil$, we have $\{M \leq t\} = \{m - t \leq N_2 \leq t\}$, a symmetric central interval event. For binomial distributions with parameter $q \geq 1/2$, the probability of such a central interval is nonincreasing in $q$. Hence, for all $t$, $\Pr(M \leq t) \geq \Pr(M' \leq t)$, because $q' \geq q$. This implies that $M'$ first-order stochastically dominates $M$. By Strassen's coupling theorem (Strassen, 1965), there exists a coupling with $M \leq M'$ almost surely. Combining with the shared $k$ yields the desired coupling of $(\vec{N}, \vec{N}')$. □

## B.6. Proof of Lemma B.8

*Proof.* Fixing an $i \in [n]$, we first compute the marginal distribution of $T_i$. For any label $j \in [m]$,

$$\Pr(T_i = j) = \sum_{a \neq j} \Pr(A_i = a) \Pr(T_i = j \mid A_i = a) = \sum_{a \neq j} q_a \cdot \frac{1}{m-1} = \frac{1 - q_j}{m - 1}.$$

Therefore, for any $j \in [m]$,

$$\Pr(Z_i = j) = \alpha \Pr(X_i = j) + (1 - \alpha) \Pr(T_i = j) = \alpha p_j + (1 - \alpha) \frac{1 - q_j}{m - 1} =: r_j,$$

which shows $Z_i \sim r$. □

## B.7. Formal Definition of Monotone Updating Rule

Here, we formalize the definition of the *monotone updating rule* and provide an illustrative example. To better illustrate our results, we focus on an arbitrary round $t$ and omit the superscript. Moreover, we define a permutation-specific notion of majorization (Definition 4.1). For vectors $v, v' \in \mathbb{R}^m$ and permutations $\sigma, \sigma'$ of $[m]$, we write $v' \succeq_{\sigma', \sigma} v$ if $\sum_i^k v'_{\sigma'(i)} \geq \sum_i^k v_{\sigma(i)}$ for all $k \in [m]$.

**Definition B.6** (**Monotone Updating Rule**). Given any answer distributions $p = (p_1, (1 - p_1)e)$ and $p'$, and answer count vectors $\vec{N} = (N_1, (n - N_1)\epsilon)$ and $\vec{N}'$, for any permutations $\sigma, \sigma'$ of $\{2, \dots, m\}$ (if exist) such that $e' \succeq_{\sigma', \sigma} e$ and $\epsilon' \succeq_{\sigma', \sigma} \epsilon$, let $(p_1^+, (1 - p_1^+)e^+) := \gamma(p, \vec{N})$ and $(p_1'^+, (1 - p_1'^+)e'^+) := \gamma(p', \vec{N}')$ be the next-round answer distributions. An updating rule $\gamma$ is *monotone* if:

- it is *error entropy monotone*, i.e. $e'^+ \succeq_{\sigma', \sigma} e^+$.
- it is *accuracy monotone*: If additionally $p_1 \geq p_1'$ and $N_1/n \geq N_1'/n'$, then $p_1^+ \geq p_1'^+$.

*Example* 1 (Linear aggregation). It is easy to verify that the linear updating rule $\gamma_\mu$ is monotone, where

$$\gamma_\mu(p, \vec{N}) = \mu p + (1 - \mu)\vec{N}/n, \qquad \alpha \in [0, 1].$$

First, majorization is preserved under convex combinations, implying that error entropy-monotonicity holds. Moreover, the updated accuracy satisfies $p_1^+ = \mu p_1 + (1 - \mu)N_1/n$, which is nondecreasing in both $p_1$ and $N_1/n$.

## B.8. Proof of Proposition 4.9

*Proof.* Let $s := (1 - q)/2$ be the distribution of the compensate vote. By Lemma B.8, for $m = 3$ the mixture rule is equivalent to vanilla majority vote on $n$ i.i.d. draws from

$$r(\alpha) = \big(r_1(\alpha), r_2(\alpha), r_3(\alpha)\big) = \alpha(p_1, p_2, p_3) + (1 - \alpha)(s_1, s_2, s_3).$$

Parameterize any $x = (x_1, x_2, x_3) \in \Delta^3$ as

$$\rho(x) := x_1, \qquad u(x) := \frac{\max\{x_2, x_3\}}{1 - x_1} \in \left[\tfrac{1}{2}, 1\right].$$

When $q_1 \leq 1 - 2p_1$ and $q_2 \geq q_3$, let

$$\alpha^* = \frac{s_3 - s_2}{p_2 - p_3 + s_3 - s_2} \in [0, 1).$$

We show that $\mathrm{Acc}^{\mathrm{MV}}(r(\alpha^*); n) > \mathrm{Acc}^{\mathrm{MV}}(r(1); n)$. By Lemma B.4, it is sufficient to show that $\rho(r(\alpha^*)) > \rho(r(1))$ and $u(r(\alpha^*)) < u(r(1))$.

First, because $s_1 > p_1$, for any $\alpha < 1$, $r_1(\alpha) = \alpha p_1 + (1 - \alpha)s_1 > p_1 = r_1(1)$. Therefore, $\rho(r(\alpha^*)) > \rho(r(1))$. Second, it is easy to observe that

$$r_2(\alpha^*) = r_3(\alpha^*) = \frac{p_2 s_3 - s_2 p_2}{p_2 - p_3 + s_3 - s_2},$$

implying that $u(r(\alpha^*)) = \tfrac{1}{2} < \frac{p_2}{p_2 + p_3} = u(r(1))$. By Lemma B.4, $\mathrm{Acc}^{\mathrm{MV}}(r(\alpha^*); n) > \mathrm{Acc}^{\mathrm{MV}}(r(1); n)$.

Furthermore, by Theorem 4.6, $R(r(\alpha^*); \bar{n}) > R(r(1); \bar{n})$ for $\bar{n} \geq 2$. This completes the proof. □

## B.9. Mixture Voting Rule

To facilitate analysis, we reformulate anti-voting as a *mixture voting rule*.

**Definition B.7** (**Mixture Voting Rule**). Let $p, q \in \Delta^m$ be the distributions of votes and anti-votes, respectively, and let $\alpha \in [0, 1]$. For each $i \in [n]$,

1. Draw a vote $Z_i \sim p$ and an anti-vote $A_i \sim q$.
2. Given $A_i = a$, draw a compensate vote $T_i$ uniformly from $[m] \setminus \{a\}$.
3. Output a single vote $Z_i = X_i$ with probability $\alpha$ and $T_i$ otherwise.

The final decision is made by applying vanilla majority voting to $Z_i$.

This reinterpretation converts anti-voting into a standard majority-vote procedure applied to a mixture of two types of votes, allowing the results from previous subsections to be applied directly.[2]

**Lemma B.8** (**Vote-count Equivalence of Mixture Voting**). *Under Definition* B.7, *the output votes* $Z_1, \ldots, Z_n$ *are i.i.d. with distribution* $r(\alpha) = \alpha p + (1 - \alpha)\frac{1-q}{m-1} \in \Delta^m$.

Lemma B.8 shows that mixture voting is distributionally equivalent to vanilla majority voting on $n$ i.i.d. samples from $r$, under the same tie-breaking rule. We therefore study (i) when anti-votes help (i.e., when some $\alpha < 1$ outperforms $\alpha = 1$) and (ii) which anti-vote distribution $q$ is preferred, by analyzing majority-vote accuracy under the induced multinomial distribution $r(\alpha)$. Recall that $\mathrm{Acc}^{\mathrm{MV}}(p; n)$ is the majority-vote accuracy of $n$ i.i.d./ samples from $p$.

## B.10. Proof of Proposition B.9

*Proof.* We prove the proposition by showing that for any $\alpha' \in [0, 1]$ and any $n \geq 1$, we can find an $\alpha \in [0, 1]$ such that $\mathrm{Acc}^{\mathrm{MV}}(r(\alpha); n) \geq \mathrm{Acc}^{\mathrm{MV}}(r'(\alpha'); n)$. We apply the same notation as in Proposition 4.9. Let $s := (1 - q)/2$ and $s' := (1 - q')/2$ be the distributions of the compensate vote. We have $s_1 = s_1' < p_1$, $s_2 \leq s_3$, and $s_2 < s_2'$. Parameterize any $x = (x_1, x_2, x_3) \in \Delta^3$ as

$$\rho(x) := x_1, \qquad u(x) := \frac{\max\{x_2, x_3\}}{1 - x_1} \in \left[\tfrac{1}{2}, 1\right].$$

An important point is $\bar{\alpha} = \frac{s_3 - s_2}{p_2 - p_3 + s_3 - s_2} \in [0, 1)$, where $r_2(\bar{\alpha}) = r_3(\bar{\alpha})$ and $u(r(\bar{\alpha})) = \frac{1}{2}$.

**Case 1:** $\alpha' \in [\bar{\alpha}, 1]$. For this case, we set $\alpha = \alpha'$. This implies that $r_1(\alpha) = r_1'(\alpha')$, which in turn suggests that $\rho(r(\alpha)) = \rho(r'(\alpha'))$. Next, we show that $u(r(\alpha)) < u(r'(\alpha'))$. If so, by Lemma B.4, $\mathrm{Acc}^{\mathrm{MV}}(r(\alpha); n) \geq \mathrm{Acc}^{\mathrm{MV}}(r'(\alpha'); n)$. Because $\alpha \geq \bar{\alpha}$, it is easy to observe that

$$r_2(\alpha) \geq r_3(\alpha).$$

Moreover, because $s_2' > s_2$,

$$r_2'(\alpha') = \alpha p_2 + (1 - \alpha)s_2' > \alpha p_2 + (1 - \alpha)s_2 = r_2(\alpha).$$

Similarly, because $s_3' = 1 - s_1' - s_2' < 1 - s_1 - s_2 = s_3$,

$$r_3'(\alpha') = \alpha p_3 + (1 - \alpha)s_3' < \alpha p_3 + (1 - \alpha)s_3 = r_3(\alpha).$$

Therefore, $r_2'(\alpha') > r_2(\alpha) \geq r_3(\alpha) > r_3'(\alpha')$. Putting everything together, we have $r_1(\alpha) = r_1'(\alpha')$, $\max(r_2'(\alpha'), r_3'(\alpha')) = r_2'(\alpha') > \max(r_2(\alpha), r_3(\alpha)) = r_2(\alpha)$. This suggests that $u(r(\alpha)) < u(r'(\alpha'))$, which completes the proof for case 1.

**Case 2:** $\alpha' \in [0, \bar{\alpha})$. For this case, we set $\alpha = \bar{\alpha}$. Note that $u(r(\bar{\alpha})) = \frac{1}{2}$, which is the minimum value. It suffices to show that $\rho(r'(\alpha')) \leq \rho(r(\bar{\alpha}))$ for any $\alpha' \in [0, \bar{\alpha})$. This is straightforward because $\alpha' < \bar{\alpha}$ implies that $r_1'(\alpha') < r_1(\bar{\alpha})$. This completes the proof. $\square$

---

[2]We use mixture voting for analytical clarity. Although it is not exactly equivalent to the anti-voting in Section 3.4, the two rules are asymptotically equivalent as the number of votes grows.

### B.11. Accurate Common Mistake Predictors Benefit Anti-Voting

We next turn to settings where anti-votes are not guaranteed to be strictly better, while noting that they are always weakly beneficial since setting $\alpha = 1$ recovers the baseline. In such cases, we show that whenever anti-votes are strictly beneficial, a predictor that more accurately identifies the common mistake is preferred.

**Proposition B.9** (**Better Common-Mistake Predictors Improve Majority Accuracy**). *Suppose $m = 3$ and the vote distribution is $p = (p_1, p_2, p_3) \in \Delta^3$ with $p_2 > p_3$. Consider two anti-vote distributions $q, q' \in \Delta^3$ satisfying $q_1 = q_1' \geq 1 - 2p_1$, $q_2 \geq q_3$, and $q_2 > q_2'$. Let $r(\alpha) = \alpha p + (1 - \alpha)q$ and $r'(\alpha) = \alpha p + (1 - \alpha)q'$. Then, for any $n \geq 1$, $\max_\alpha \mathrm{Acc}^{\mathrm{MV}}(r(\alpha); n) \geq \max_\alpha \mathrm{Acc}^{\mathrm{MV}}(r'(\alpha); n)$.*

### B.12. The Optimal $\eta$ in Anti-Voting

Recall the anti-vote setting in Section 3.4. For a fixed question, we obtain $n$ i.i.d. model outputs, each consisting of an answer $Z_i \in [m]$ and an anti-vote $A_i \in [m]$, interpreted as a prediction of the most common mistake. Since the same model and prompt are used independently, the pairs $(Z_i, A_i)_{i=1}^n$ are i.i.d. draws from a joint distribution $P(Z, A)$. This joint distribution determines aggregation behavior.

To understand how the anti-vote weight $\eta$ should be chosen, it is helpful to consider a geometric view of $P(Z, A)$ (Figure 7 presents a three-choice example). Each answer $j$ corresponds to a point $(p_j, q_j)$ in the plane, where the x-axis represents the expected fraction of answer votes and the y-axis represents the expected fraction of anti-votes. When $n \to \infty$, empirical frequencies concentrate at these points almost surely, so aggregation reduces to a deterministic comparison of linear scores. In this limit, standard majority vote ($\eta = 0$) selects the answer with the largest $p_j$, while very large $\eta$ selects the answer with the smallest $q_j$. Intermediate values of $\eta$ interpolate between these two extremes by tilting the maximization direction.

**Case $m = 3$.**  Suppose there are three possible answers, with ground truth $y$ and two incorrect answers $b$ and $c$. In the $(p, q)$-plane, the three answers correspond to points $(p_y, q_y)$, $(p_b, q_b)$, and $(p_c, q_c)$. As $n \to \infty$, the aggregation rule selects the answer maximizing $p_j - \eta q_j$, which geometrically corresponds to projecting these points onto a direction with slope $-\eta$.

If $(p_y, q_y)$ lies strictly between $(p_b, q_b)$ and $(p_c, q_c)$ along some direction, then there exists a nonempty interval of $\eta$ values for which $y$ strictly dominates both incorrect answers. More precisely, provided the denominators are nonzero, any

$$\eta \in \left( \frac{p_b - p_y}{q_b - q_y}, \; \frac{p_y - p_c}{q_y - q_c} \right)$$

ensures that $p_y - \eta q_y > p_b - \eta q_b$ and $p_y - \eta q_y > p_c - \eta q_c$, so that the aggregation outcome equals $y$ with probability one in the infinite-sample limit.

When $n < \infty$, the realized vote counts fluctuate around their expectations. Although an exact characterization of the optimal $\eta$ is analytically intractable, the same geometry provides useful intuition. For a fixed $\eta$, the probability of correct aggregation depends on the margin between the score of $y$ and the largest competing score. Larger margins imply greater robustness to sampling noise. Consequently, a natural heuristic is to choose $\eta$ to maximize the separation between $y$ and its strongest competitor. In the two-error case, this corresponds to selecting the direction orthogonal to the line connecting $(p_b, q_b)$ and $(p_c, q_c)$, yielding the heuristic choice

$$\eta^\star = \frac{p_b - p_c}{q_b - q_c}.$$

While this choice is approximate, it becomes increasingly accurate as $n$ grows.

**Case $m \geq 3$.**  When there are more than two incorrect answers, the same reasoning generalizes. Consider the set of points $\{(p_j, q_j) : j \neq y\}$ corresponding to all incorrect answers. Among these, only points on the Pareto frontier are relevant competitors: these are points that cannot be simultaneously improved by increasing $p_j$ and decreasing $q_j$. Equivalently, they form the outer boundary of the convex hull of the incorrect answers.

For any candidate direction determined by $\eta$, aggregation compares the projection of $(p_y, q_y)$ against the maximum projection over this frontier. The most robust choice of $\eta$ is therefore the one that maximizes the worst-case separation between $y$ and all incorrect answers. Formally, let $\vec{d}_j$ denote the vector from a frontier point $j$ to $y$, and let

$$\vec{e}_\eta = \frac{1}{\sqrt{1 + \eta^2}} (1, -\eta)$$

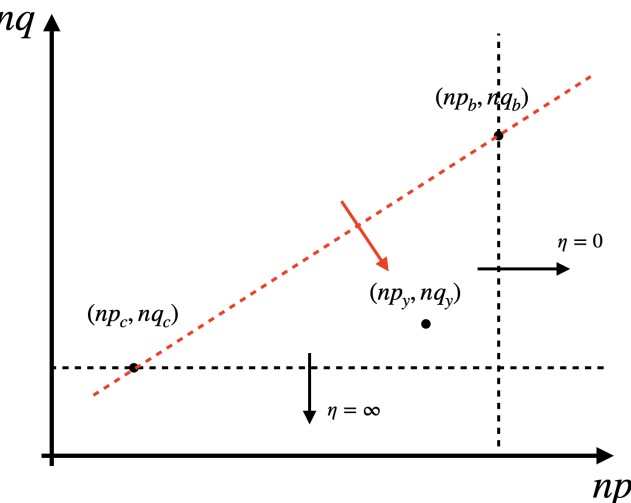

*Figure 7.* An illustration of the optimal majority vote with prediction of mistakes when $m = 3$.

be the unit direction associated with $\eta$. The optimal direction solves

$$\eta^{\star} \;=\; \arg\max_{\eta} \; \min_{j} \; \langle \vec{d}_j, \vec{e}_\eta \rangle,$$

where the minimum ranges over all frontier points. This choice maximizes the smallest margin between $y$ and any competing answer, balancing answer accuracy and anti-vote information to achieve the most robust aggregation.

## C. Additional Experimental Details

**Checkpoint Selection.** For each training run, we select a checkpoint that has highest average reward on 1000 randomly chosen samples from the unfiltered train set. These samples effectively serve as the validation set.

**Temperature Setting.** For all inference generations, we use the temperature that yields the best performance for each model. Our training results show that controlling error diversity provides performance gains beyond those achievable by increasing temperature alone. In practice, temperature tuning has mixed effects across models: while some models benefit from higher temperatures, others suffer from reduced coherence and accuracy. For Gemma-2 we found this to be temperature 1.0, for the Qwen family we found this to be temperature 1.2 and for llama we found this to be 0.8. Temperature was selected for majority vote accuracy.

**Data Construction.** Training data are generated offline by sampling tasks $(x, y)$ from a fixed dataset and estimating $\tilde{p}$ using the current or a frozen policy. By avoiding on-policy rollouts during training, this approach substantially reduces computational overhead and improves training throughput.

*Training examples:* For vanilla GRPO we train directly on round 0 prompts (one example per question). For the other methods, Reflect-GRPO, Trained Dist-Reflection, and D-FUSEr we create the data by generating 4 sample trajectories per question. For our method we do not use round 0 prompts.

*Data filtering:* After the construction above, we filter the data according to difficulty. Any question that the model can answer more than 90% accuracy or less than 10% accuracy is dropped (to avoid settings with 0 advantage value). For fairness we apply the same filtering to all methods.

**Training Configurations** All training methods are trained using 1 epoch. For all experiments we use using GRPO training we use a learning rate of $1e{-}6$ with $\beta = 0.0$ and a group size of 16.

In table 4 the average total GPU hours averaged across all datasets. Total hours is calculated as num_GPUs * hours

*Table 4.* Average GPU hours per method, averaged across datasets.

| Method | Qwen-14B | Qwen-7B | Qwen-3B | Gemma-2B | Llama-8B |
|---|---|---|---|---|---|
| GRPO | 160 | 120 | 112 | 128 | 136 |
| Reflect-GRPO | 224 | 192 | 160 | 184 | 200 |
| Dist-Reflect | 240 | 208 | 184 | 208 | 224 |
| D-FUSEr | 248 | 200 | 176 | 208 | 184 |

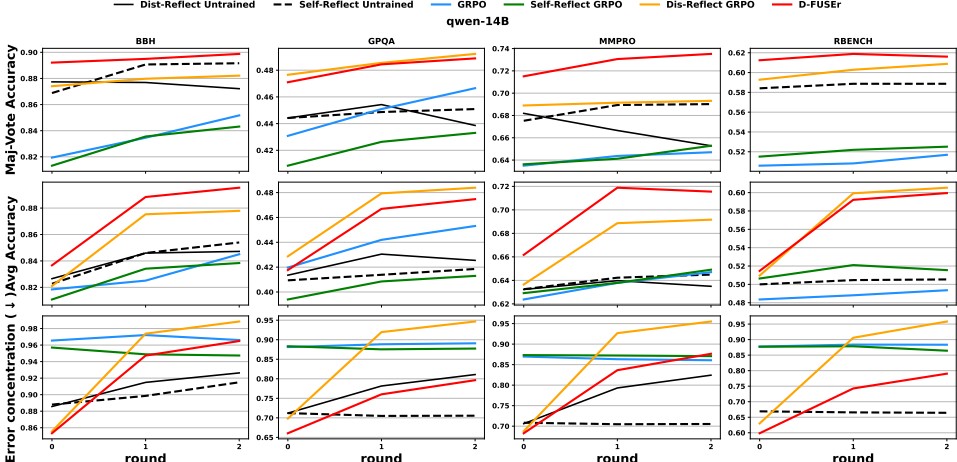

*Figure 8.* Per round performance of Qwen-2.5-14B.

## D. Additional Experimental Results for $r_{\text{D-FUSEr}}$

| | OOD | | | | In Distribution | |
|---|---|---|---|---|---|---|
| | BBH | GPQA | MMPRO | RBENCH | BBH | MMPRO |
| — Untrained — | | | | | | |
| Zeroshot | $\underline{48.0}_{\pm0.8}$ | $27.1_{\pm1.1}$ | $31.0_{\pm0.4}$ | $16.6_{\pm0.7}$ | $48.0_{\pm1.0}$ | $31.0_{\pm0.4}$ |
| Self-Reflect | $\underline{46.8}_{\pm0.8}$ | $27.2_{\pm1.1}$ | $34.9_{\pm0.5}$ | $23.4_{\pm0.8}$ | $46.8_{\pm0.9}$ | $34.9_{\pm0.4}$ |
| Dis-Reflect | $44.4_{\pm1.0}$ | $26.1_{\pm1.2}$ | $31.3_{\pm0.4}$ | $25.3_{\pm0.8}$ | $44.4_{\pm0.9}$ | $31.3_{\pm0.5}$ |
| — Trained — | | | | | | |
| GRPO | $46.8_{\pm0.9}$ | $29.6_{\pm1.2}$ | $\underline{35.8}_{\pm0.5}$ | $41.1_{\pm1.0}$ | $57.5_{\pm0.8}$ | $\underline{40.9}_{\pm0.5}$ |
| Reflect-GRPO | $47.1_{\pm0.8}$ | $\underline{31.9}_{\pm1.3}$ | $34.2_{\pm0.4}$ | $37.5_{\pm0.9}$ | $\underline{59.0}_{\pm0.9}$ | $40.5_{\pm0.5}$ |
| Dist-Reflect | $47.9_{\pm0.9}$ | $\underline{31.9}_{\pm1.3}$ | $32.3_{\pm0.5}$ | $\underline{41.6}_{\pm1.0}$ | $56.0_{\pm0.9}$ | $39.6_{\pm0.5}$ |
| D-FUSEr (ours) | $\mathbf{51.2}_{\pm0.9}$ | $\mathbf{37.7}_{\pm1.4}$ | $\mathbf{39.6}_{\pm0.5}$ | $\mathbf{42.3}_{\pm1.0}$ | $\mathbf{65.2}_{\pm0.8}$ | $\mathbf{43.0}_{\pm0.5}$ |

*Table 5.* Majority-vote accuracy (%) for gemma-2B. Best is bold; second-best is underlined (per column).

### D.1. Per-round Performance Comparison

## E. Additional Experimental Results on Anti-votes.

We present our experimental results on anti-votes for all datasets in Figure 16.

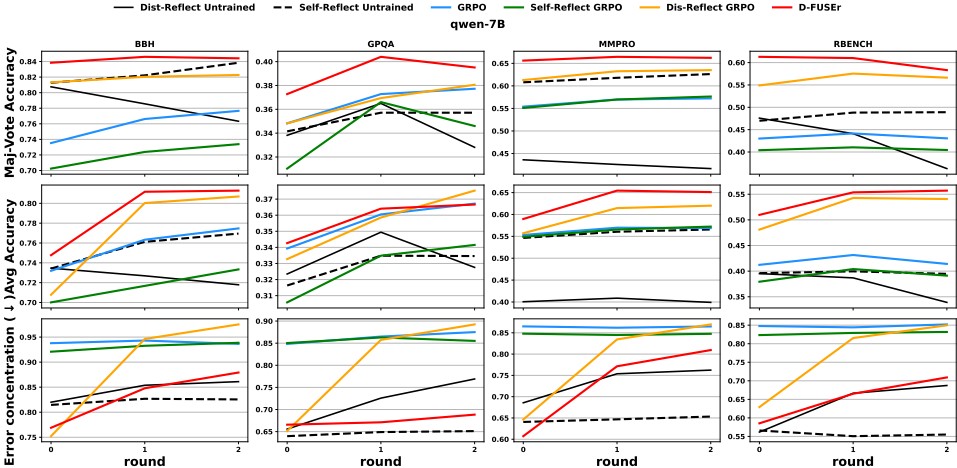

*Figure 9.* Per round performance of Qwen-2.5-7B.

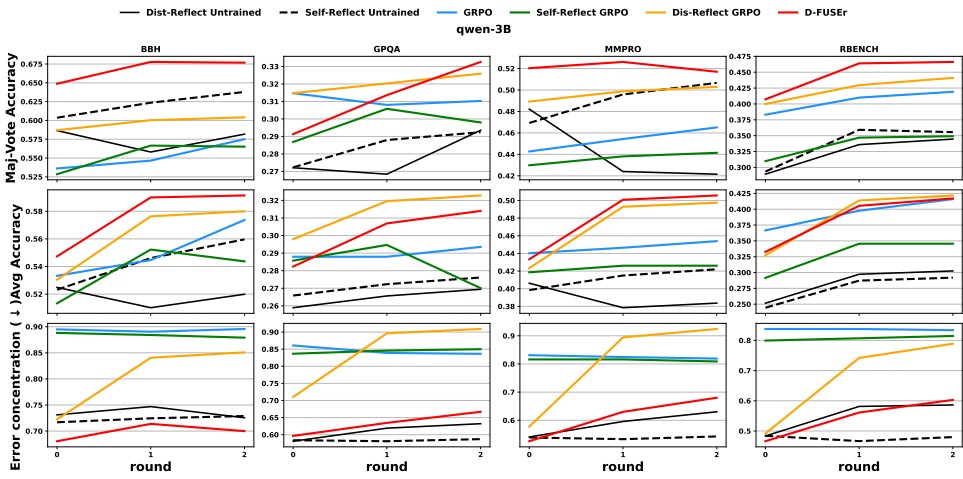

*Figure 10.* Per round performance of Qwen-2.5-3B.

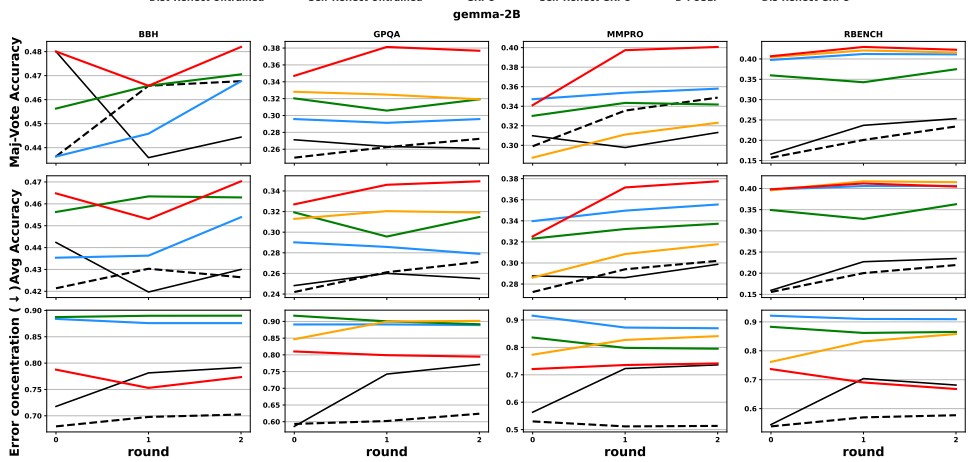

*Figure 11.* Per round performance of Gemma-2B.

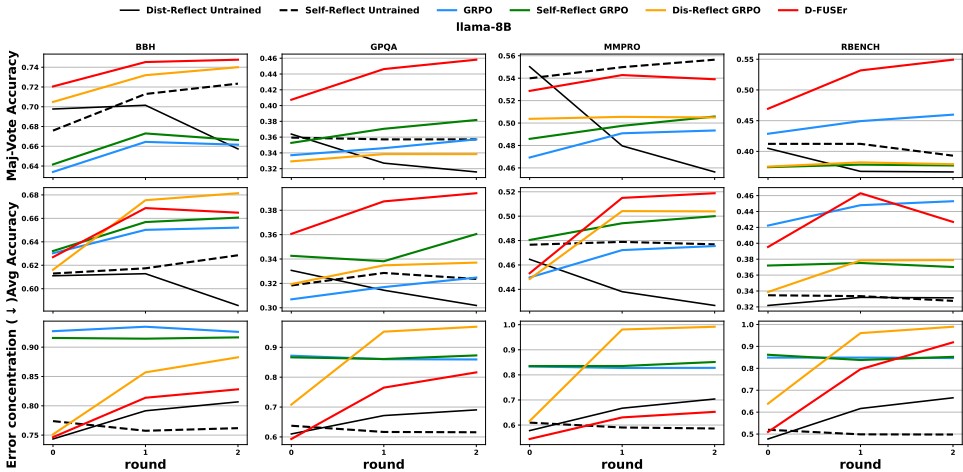

*Figure 12.* Per round performance of Llama-3.1-8B.

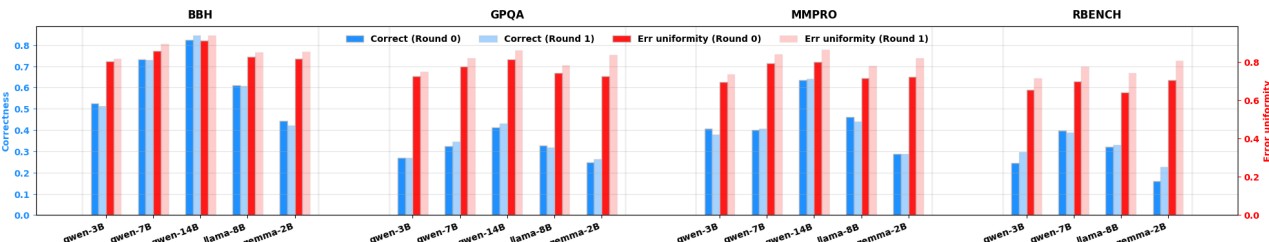

*Figure 13.* Correctness and error uniformity from roud 0 to round 1.

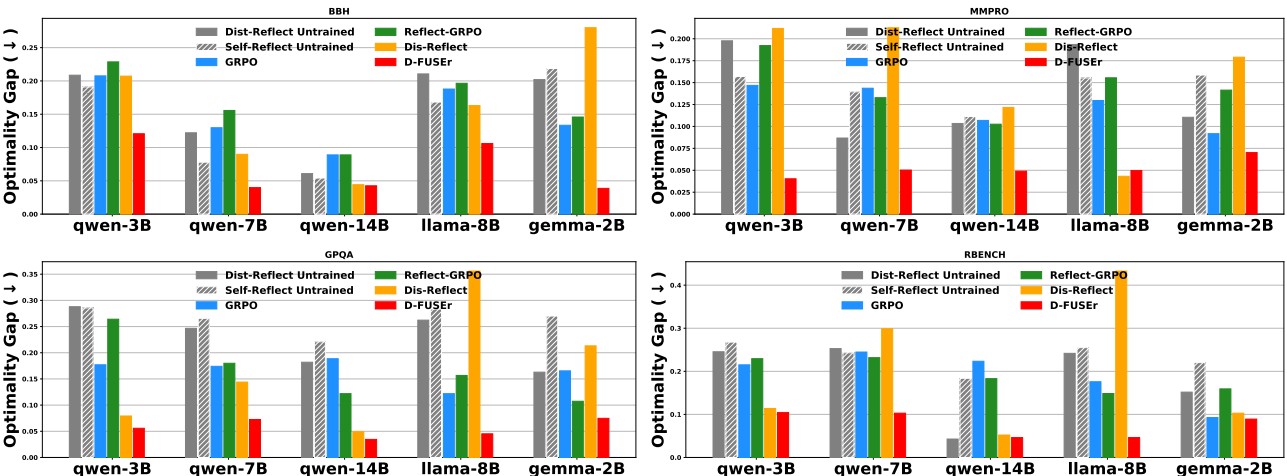

*Figure 14.* Majority vote optimality gap for all methods and datasets.

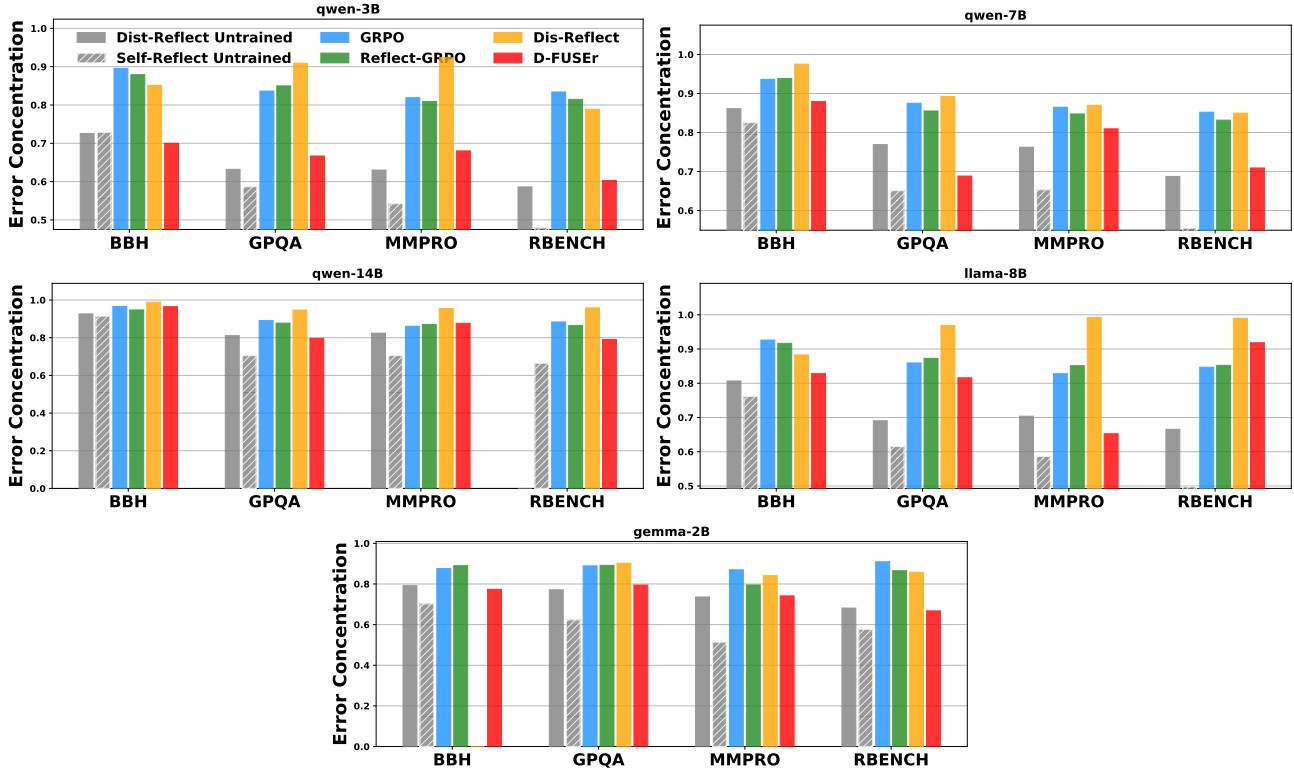

*Figure 15.* Error concentration for all methods and datasets

|  | OOD | | | | In Distribution | |
|---|---|---|---|---|---|---|
|  | BBH | GPQA | MMPRO | RBENCH | BBH | MMPRO |
| — Untrained — | | | | | | |
| Zeroshot | $69.8_{\pm 0.9}$ | $36.4_{\pm 1.4}$ | $\underline{55.0}_{\pm 0.5}$ | $40.5_{\pm 0.9}$ | $69.8_{\pm 0.8}$ | $55.0_{\pm 0.5}$ |
| Self-Reflect | $\underline{72.3}_{\pm 0.7}$ | $35.7_{\pm 1.3}$ | $\mathbf{55.6}_{\pm 0.5}$ | $39.3_{\pm 0.9}$ | $72.3_{\pm 0.8}$ | $\underline{55.6}_{\pm 0.5}$ |
| Dis-Reflect | $65.7_{\pm 0.9}$ | $31.6_{\pm 1.4}$ | $45.6_{\pm 0.5}$ | $36.7_{\pm 0.9}$ | $65.7_{\pm 0.8}$ | $45.6_{\pm 0.5}$ |
| — Trained — | | | | | | |
| GRPO | $66.2_{\pm 0.8}$ | $35.7_{\pm 1.3}$ | $49.3_{\pm 0.5}$ | $\underline{46.0}_{\pm 1.0}$ | $\mathbf{77.2}_{\pm 0.7}$ | $\mathbf{59.4}_{\pm 0.4}$ |
| Reflect-GRPO | $64.5_{\pm 0.8}$ | $\underline{38.2}_{\pm 1.2}$ | $50.6_{\pm 0.5}$ | $37.7_{\pm 0.9}$ | $66.6_{\pm 0.9}$ | $52.7_{\pm 0.5}$ |
| Dist-Reflect | $71.5_{\pm 0.9}$ | $33.8_{\pm 1.3}$ | $48.5_{\pm 0.5}$ | $37.9_{\pm 0.9}$ | $74.0_{\pm 0.8}$ | $50.5_{\pm 0.4}$ |
| D-FUSEr (ours) | $\mathbf{73.4}_{\pm 0.8}$ | $\mathbf{45.8}_{\pm 1.3}$ | $52.2_{\pm 0.5}$ | $\mathbf{54.9}_{\pm 0.9}$ | $\underline{74.8}_{\pm 0.8}$ | $53.9_{\pm 0.5}$ |

*Table 6.* Majority-vote accuracy (%) for llama-8B. Best is bold; second-best is underlined (per column).

| | OOD | | | | In Distribution | |
|---|---|---|---|---|---|---|
| | BBH | GPQA | MMPRO | RBENCH | BBH | MMPRO |
| | — Untrained — | | | | | |
| Zeroshot | $58.7_{\pm0.9}$ | $27.2_{\pm1.2}$ | $48.2_{\pm0.5}$ | $28.9_{\pm0.9}$ | $58.7_{\pm1.0}$ | $48.2_{\pm0.5}$ |
| Self-reflect | $\underline{63.8}_{\pm0.8}$ | $29.2_{\pm1.3}$ | $\underline{50.7}_{\pm0.5}$ | $35.6_{\pm0.9}$ | $63.8_{\pm0.9}$ | $50.7_{\pm0.5}$ |
| Dis-reflect | $58.2_{\pm0.8}$ | $29.4_{\pm1.4}$ | $42.2_{\pm0.4}$ | $34.5_{\pm0.8}$ | $58.2_{\pm0.9}$ | $42.2_{\pm0.5}$ |
| | — Trained — | | | | | |
| GRPO | $57.5_{\pm0.9}$ | $31.0_{\pm1.3}$ | $46.5_{\pm0.5}$ | $41.9_{\pm1.0}$ | $\underline{69.6}_{\pm0.9}$ | $\mathbf{54.6}_{\pm0.5}$ |
| Reflect-GRPO | $56.5_{\pm0.9}$ | $29.8_{\pm1.2}$ | $44.1_{\pm0.4}$ | $34.9_{\pm0.9}$ | $64.0_{\pm0.9}$ | $52.2_{\pm0.5}$ |
| Dist-Reflect | $60.4_{\pm1.0}$ | $\underline{32.6}_{\pm1.4}$ | $48.5_{\pm0.4}$ | $\underline{44.1}_{\pm1.1}$ | $62.8_{\pm0.9}$ | $50.3_{\pm0.5}$ |
| D-FUSEr (ours) | $\mathbf{67.7}_{\pm0.8}$ | $\mathbf{35.3}_{\pm1.2}$ | $\mathbf{51.7}_{\pm0.5}$ | $\mathbf{46.6}_{\pm1.1}$ | $\mathbf{72.0}_{\pm0.8}$ | $\underline{52.6}_{\pm0.5}$ |

*Table 7.* Majority-vote accuracy (%) for Qwen-2.5-3B. Best is bold; second-best is underlined (per column).

| | OOD | | | | In Distribution | |
|---|---|---|---|---|---|---|
| | BBH | GPQA | MMPRO | RBENCH | BBH | MMPRO |
| | — Untrained — | | | | | |
| Zeroshot | $80.8_{\pm0.7}$ | $33.8_{\pm1.2}$ | $43.6_{\pm0.4}$ | $47.6_{\pm1.1}$ | $80.8_{\pm0.7}$ | $43.6_{\pm0.5}$ |
| Self-reflect | $\underline{83.8}_{\pm0.6}$ | $35.7_{\pm1.2}$ | $\mathbf{62.6}_{\pm0.5}$ | $48.9_{\pm0.9}$ | $83.8_{\pm0.6}$ | $62.6_{\pm0.5}$ |
| Dis-reflect | $76.3_{\pm0.8}$ | $32.8_{\pm1.4}$ | $41.7_{\pm0.4}$ | $36.3_{\pm1.0}$ | $76.3_{\pm0.7}$ | $41.7_{\pm0.5}$ |
| | — Trained — | | | | | |
| GRPO | $77.7_{\pm0.7}$ | $37.7_{\pm1.3}$ | $57.2_{\pm0.4}$ | $43.1_{\pm0.9}$ | $85.2_{\pm0.6}$ | $62.6_{\pm0.5}$ |
| Reflect-GRPO | $73.4_{\pm0.8}$ | $34.6_{\pm1.3}$ | $57.6_{\pm0.4}$ | $40.4_{\pm1.0}$ | $83.6_{\pm0.7}$ | $62.0_{\pm0.5}$ |
| Dist-Reflect | $82.3_{\pm0.7}$ | $\underline{38.1}_{\pm1.3}$ | $60.8_{\pm0.5}$ | $\underline{56.6}_{\pm0.9}$ | $\mathbf{90.9}_{\pm0.5}$ | $\underline{63.5}_{\pm0.5}$ |
| D-FUSEr (ours) | $\mathbf{85.4}_{\pm0.6}$ | $\mathbf{41.5}_{\pm1.4}$ | $\underline{61.9}_{\pm0.5}$ | $\mathbf{59.3}_{\pm0.9}$ | $\underline{87.5}_{\pm0.6}$ | $\mathbf{66.2}_{\pm0.5}$ |

*Table 8.* Majority-vote accuracy (%) for qwen-7B. Best is bold; second-best is underlined (per column).

| | OOD | | | | In Distribution | |
|---|---|---|---|---|---|---|
| | BBH | GPQA | MMPRO | RBENCH | BBH | MMPRO |
| | — Untrained — | | | | | |
| Zeroshot | $87.7_{\pm0.6}$ | $44.4_{\pm1.3}$ | $68.2_{\pm0.5}$ | $58.4_{\pm0.9}$ | $87.7_{\pm0.6}$ | $68.2_{\pm0.5}$ |
| Self-reflect | $\underline{89.2}_{\pm0.6}$ | $45.1_{\pm1.4}$ | $69.0_{\pm0.5}$ | $58.9_{\pm1.0}$ | $89.2_{\pm0.6}$ | $69.0_{\pm0.4}$ |
| Dis-reflect | $87.2_{\pm0.6}$ | $43.9_{\pm1.4}$ | $65.3_{\pm0.5}$ | $56.1_{\pm1.0}$ | $87.2_{\pm0.6}$ | $65.3_{\pm0.5}$ |
| | — Trained — | | | | | |
| GRPO | $85.2_{\pm0.6}$ | $46.7_{\pm1.3}$ | $64.7_{\pm0.5}$ | $51.7_{\pm1.1}$ | $89.9_{\pm0.5}$ | $\underline{69.5}_{\pm0.4}$ |
| Reflect-GRPO | $84.3_{\pm0.6}$ | $43.3_{\pm1.3}$ | $65.3_{\pm0.5}$ | $52.5_{\pm1.0}$ | $90.1_{\pm0.5}$ | $69.2_{\pm0.5}$ |
| Dist-Reflect | $88.2_{\pm0.6}$ | $\mathbf{49.2}_{\pm1.5}$ | $\underline{70.1}_{\pm0.4}$ | $\underline{60.9}_{\pm1.0}$ | $\mathbf{92.3}_{\pm0.5}$ | $69.3_{\pm0.5}$ |
| D-FUSEr (ours) | $\mathbf{91.9}_{\pm0.4}$ | $\underline{48.9}_{\pm1.4}$ | $\mathbf{73.5}_{\pm0.4}$ | $\mathbf{65.6}_{\pm0.9}$ | $\underline{91.4}_{\pm0.5}$ | $\mathbf{75.7}_{\pm0.4}$ |

*Table 9.* Majority-vote accuracy (%) for qwen-14B. Best is bold; second-best is underlined (per column).

| | OOD | | | | In Distribution | |
|---|---|---|---|---|---|---|
| | BBH | GPQA | MMPRO | RBENCH | BBH | MMPRO |
| **Qwen-3B** | | | | | | |
| GRPO | $57.4_{\pm1.3}$ | $29.4_{\pm1.8}$ | $\underline{45.4}_{\pm0.7}$ | $41.6_{\pm1.4}$ | $\underline{60.7}_{\pm1.2}$ | $46.9_{\pm0.7}$ |
| Reflect-GRPO | $54.4_{\pm1.3}$ | $27.0_{\pm1.7}$ | $42.6_{\pm0.7}$ | $34.6_{\pm1.3}$ | $56.3_{\pm1.3}$ | $43.2_{\pm0.7}$ |
| Dist-Reflect | $\underline{58.0}_{\pm1.2}$ | $\mathbf{32.3}_{\pm1.8}$ | $42.1_{\pm0.7}$ | $\mathbf{42.2}_{\pm1.4}$ | $60.6_{\pm1.2}$ | $\underline{49.7}_{\pm0.7}$ |
| D-FUSEr (ours) | $\mathbf{59.1}_{\pm1.2}$ | $\underline{31.4}_{\pm1.8}$ | $\mathbf{49.3}_{\pm0.7}$ | $\underline{41.7}_{\pm1.4}$ | $\mathbf{63.9}_{\pm1.2}$ | $\mathbf{50.6}_{\pm0.7}$ |
| **Qwen-7B** | | | | | | |
| GRPO | $77.5_{\pm1.1}$ | $\underline{36.7}_{\pm1.9}$ | $56.8_{\pm0.7}$ | $41.4_{\pm1.4}$ | $79.0_{\pm1.0}$ | $56.9_{\pm0.7}$ |
| Reflect-GRPO | $73.3_{\pm1.1}$ | $34.2_{\pm1.9}$ | $56.5_{\pm0.7}$ | $39.1_{\pm1.4}$ | $76.6_{\pm1.1}$ | $57.2_{\pm0.7}$ |
| Dist-Reflect | $\underline{80.7}_{\pm1.0}$ | $\mathbf{37.5}_{\pm1.9}$ | $\mathbf{60.8}_{\pm0.7}$ | $\underline{54.1}_{\pm1.4}$ | $\mathbf{87.6}_{\pm0.8}$ | $62.0_{\pm0.7}$ |
| D-FUSEr (ours) | $\mathbf{81.2}_{\pm1.0}$ | $36.7_{\pm1.9}$ | $\underline{59.1}_{\pm0.7}$ | $\mathbf{55.7}_{\pm1.4}$ | $\underline{81.3}_{\pm1.0}$ | $\mathbf{65.1}_{\pm0.7}$ |
| **Qwen-14B** | | | | | | |
| GRPO | $84.5_{\pm0.9}$ | $45.3_{\pm2.0}$ | $64.7_{\pm0.7}$ | $49.4_{\pm1.4}$ | $85.8_{\pm0.9}$ | $65.0_{\pm0.7}$ |
| Reflect-GRPO | $83.8_{\pm0.9}$ | $41.3_{\pm1.9}$ | $64.9_{\pm0.7}$ | $51.6_{\pm1.4}$ | $86.0_{\pm0.9}$ | $65.1_{\pm0.7}$ |
| Dist-Reflect | $\underline{87.8}_{\pm0.8}$ | $\mathbf{48.4}_{\pm2.0}$ | $66.2_{\pm0.7}$ | $\mathbf{60.5}_{\pm1.4}$ | $\mathbf{91.5}_{\pm0.7}$ | $\underline{69.2}_{\pm0.6}$ |
| D-FUSEr (ours) | $\mathbf{89.4}_{\pm0.8}$ | $\underline{47.5}_{\pm2.0}$ | $\mathbf{69.8}_{\pm0.6}$ | $\underline{59.9}_{\pm1.4}$ | $\underline{89.5}_{\pm0.8}$ | $\mathbf{71.7}_{\pm0.6}$ |
| **Gemma-2B** | | | | | | |
| GRPO | $45.4_{\pm1.3}$ | $27.9_{\pm1.8}$ | $\underline{35.5}_{\pm0.7}$ | $40.6_{\pm1.4}$ | $55.4_{\pm1.3}$ | $37.1_{\pm0.7}$ |
| Reflect-GRPO | $46.3_{\pm1.3}$ | $31.5_{\pm1.8}$ | $33.7_{\pm0.7}$ | $36.3_{\pm1.3}$ | $54.5_{\pm1.3}$ | $37.7_{\pm0.7}$ |
| Dist-Reflect | $\underline{46.8}_{\pm1.3}$ | $31.9_{\pm1.8}$ | $31.8_{\pm0.6}$ | $\mathbf{41.5}_{\pm1.4}$ | $55.2_{\pm1.3}$ | $\underline{38.5}_{\pm0.7}$ |
| D-FUSEr (ours) | $\mathbf{47.0}_{\pm1.3}$ | $\mathbf{34.9}_{\pm1.9}$ | $\mathbf{37.6}_{\pm0.7}$ | $40.5_{\pm1.4}$ | $\mathbf{61.5}_{\pm1.2}$ | $\mathbf{41.4}_{\pm0.7}$ |
| **Llama-8B** | | | | | | |
| GRPO | $\underline{65.2}_{\pm1.2}$ | $32.5_{\pm1.8}$ | $\underline{47.5}_{\pm0.7}$ | $\mathbf{45.3}_{\pm1.4}$ | $\mathbf{70.8}_{\pm1.2}$ | $\mathbf{52.9}_{\pm0.7}$ |
| Reflect-GRPO | $59.2_{\pm1.2}$ | $36.0_{\pm1.9}$ | $46.3_{\pm0.7}$ | $37.0_{\pm1.3}$ | $66.1_{\pm1.2}$ | $50.0_{\pm0.7}$ |
| Dist-Reflect | $\mathbf{69.8}_{\pm1.2}$ | $33.7_{\pm1.9}$ | $37.0_{\pm0.7}$ | $37.9_{\pm1.3}$ | $\underline{68.1}_{\pm1.2}$ | $50.4_{\pm0.7}$ |
| D-FUSEr (ours) | $62.2_{\pm1.2}$ | $\mathbf{39.4}_{\pm1.9}$ | $\mathbf{50.6}_{\pm0.7}$ | $\underline{42.7}_{\pm1.4}$ | $66.5_{\pm1.2}$ | $\underline{51.9}_{\pm0.7}$ |

*Table 10.* Average accuracy (%) at the final round. Best is bold and second-best is underlined *within each base-model block* (per column).

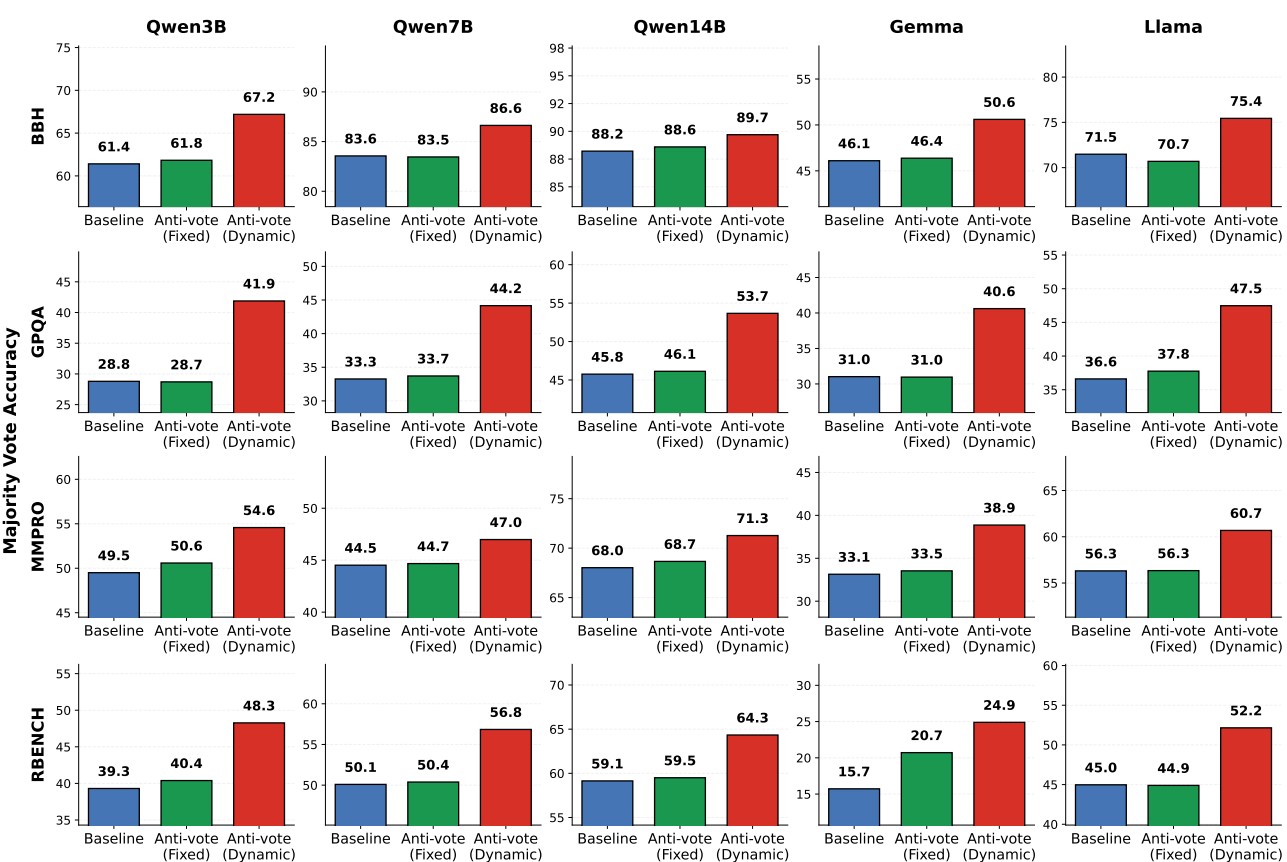

*Figure 16.* Experimental results on anti-votes for all datasets.

## F. Additional Results: Anti-vote Accuracy

Table 11 shows the anti-vote accuracy for predicting the dominant mistake.

|  | Qwen3B | Qwen7B | Qwen14B | Gemma | Llama |
|---|---|---|---|---|---|
| BBH | 32.6 | 26.5 | 26.0 | 21.8 | 38.9 |
| GPQA | 31.0 | 26.1 | 28.8 | 17.4 | 30.4 |
| MMPRO | 58.0 | 57.1 | 18.2 | 50.9 | 62.1 |
| RBENCH | 39.3 | 61.7 | 35.3 | 85.8 | 47.9 |

*Table 11.* Anti-vote accuracy (%) for predicting the dominant mistake.

## G. Prompt Examples

Here we provide prompt examples used in our experiments.

**Round 0 or zeroshot prompt:**

---

Round 0 or zeroshot prompt

**Instruction.** Please answer the following multiple choice question as accurately as possible. Make sure to first define the premises of the questions, and make sure that the answer is consistent with the premises. To answer the question, you must think step-by-step and provide detailed reasoning for your steps. You must then give your final answer as a letter by saying "Final Answer:" at the end of your response.

**Question.** Today, Emily went to the soccer field. Between what times could they have gone?

We Know that

- Emily woke up at 10am.

- Linda saw Emily waiting at the train station from 10am to 12pm.

- Ashley saw Emily fixing their computer at the electronic store from 12pm to 1pm.

- John saw Emily walking towards the Statue of Liberty from 1pm to 2pm.

- The soccer field was closed after 8pm.

**Options.**

(A) 12pm to 1pm

(B) 10am to 12pm

(C) 2pm to 8pm

(D) 1pm to 2pm

---

**Self-Reflection Prompt:**

---

Self-reflection Prompt

You previously provided an initial answer to a multiple choice question. Here is your response:

**Your Response.** To answer this question, let's analyze the given information step-by-step and determine the possible time intervals during which Emily could have gone to the soccer field. 1. **Emily's Waking Time**, Emily woke up at 10am. 2. **Linda's Observation**, Linda saw Emily waiting at the train station from 10am to 12pm. 3. **Ashley's Observation**, Ashley saw Emily fixing their computer at the electronic store from 12pm to 1pm. 4.

---

**John's Observation**, John saw Emily walking towards the Statue of Liberty from 1pm to 2pm. 5. **Soccer Field Closure**, The soccer field was closed after 8pm. Let's identify the time slots during which Emily was not accounted for, - From 10am to 12pm, Emily was at the train station. - From 12pm to 1pm, Emily was at the electronic store. - From 1pm to 2pm, Emily was walking towards the Statue of Liberty. Therefore, the time slots when Emily was not accounted for are, - 10am to 12pm - 12pm to 1pm - 1pm to 2pm Given that the soccer field was closed after 8pm, Emily could have gone to the soccer field before 8pm. The only time slot within these constraints is from 2pm to 8pm, as all other times are either before Emily was last seen or after the soccer field was closed. Final Answer, (C) 2pm to 8pm

**Instruction.** You should reflect on your initial answer and provide a revised answer to the following multiple choice question. Make sure to first define the premises of the question, and ensure the answer is consistent with those premises. To answer the question, think step-by-step and provide detailed reasoning. You must then give your final answer as a letter by saying "Final Answer:" at the end of your response.

**Question.** Today, Emily went to the soccer field. Between what times could they have gone?

We Know that

- Emily woke up at 10am.

- Linda saw Emily waiting at the train station from 10am to 12pm.

- Ashley saw Emily fixing their computer at the electronic store from 12pm to 1pm.

- John saw Emily walking towards the Statue of Liberty from 1pm to 2pm.

- The soccer field was closed after 8pm.

**Options.**

(A) 12pm to 1pm

(B) 10am to 12pm

(C) 2pm to 8pm

(D) 1pm to 2pm

---

You and several other models have provided initial answers to a multiple choice question. Because there are many responses, you will be shown (i) the distribution of predicted answers (top-3), and (ii) one example response for each of those answers.

**Predicted answer distribution (top-3).**

- Answer **C** was given 17 times out of 20.

- Answer **D** was given 2 times out of 20.

- Answer **B** was given 1 time out of 20.

**Example responses (one per answer above).**

**(Answer: C).**

Let's analyze the question step-by-step to determine when Anthony could have gone to the dance studio. 1. Anthony woke up at 8am. 2. William saw Anthony driving to the water park from 10am to 12pm. 3. David saw Anthony sitting on a rooftop from 12pm to 1pm. 4. Richard saw Anthony buying a phone at the electronics store from 1pm to 4pm. 5. The dance studio was closed after 4pm. Now, we need to identify a window of time when Anthony could have gone to the dance studio without contradicting any of the given information. ... Final Answer, (C)

**(Answer: D).**

Let's analyze the given information step-by-step to determine the possible times Anthony could have gone to the dance studio. 1. Anthony woke up at 8am. 2. William saw Anthony driving to the water park from 10am to 12pm. 3. David saw Anthony sitting on a rooftop from 12pm to 1pm. 4. Richard saw Anthony buying a phone at the electronics store from 1pm to 4pm. 5. The dance studio was closed after 4pm. ... Final Answer, (D) 10am to 12pm

**(Answer: B).**

To determine the correct time Anthony could have gone to the dance studio, let's analyze the given information step-by-step. 1. Anthony woke up at 8am. 2. William saw Anthony driving to the water park from 10am to 12pm. 3. David saw Anthony sitting on a rooftop from 12pm to 1pm. 4. Richard saw Anthony buying a phone at the electronics store from 1pm to 4pm. 5. The dance studio was closed after 4pm. ... Final Answer, (B)

Revision instruction You should take these answers into consideration when providing your revised answer to the following multiple choice question. Since the final answer will be aggregated via majority vote, it is important that you consider the distribution of answers given by other models and avoid echoing common mistakes. Remember that your initial answer or the partners' answers might be incorrect.
Make sure to first define the premises of the question, and ensure the answer is consistent with those premises. To answer the question, think step-by-step and provide detailed reasoning. You must then give your final answer as a letter by saying "Final Answer:" at the end of your response.

**Question.** Today, Anthony went to the dance studio. Between what times could they have gone?

**Facts.**

- Anthony woke up at 8am.

- William saw Anthony driving to the water park from 10am to 12pm.

- David saw Anthony sitting on a rooftop from 12pm to 1pm.

- Richard saw Anthony buying a phone at the electronics store from 1pm to 4pm.

- The dance studio was closed after 4pm.

**Options.**

(A) 12pm to 1pm

(B) 1pm to 4pm

(C) 8am to 10am

(D) 10am to 12pm

We further provide prompt examples for our anti-vote experiments below:

**Anti-vote Round0 Prompt:**

> Anti-vote Round0 Prompt
>
> Your goal is to predict the most common incorrect answer to a multiple-choice question by analyzing the kinds of mistakes large language models like you are likely to make. Consider each option carefully and analyze what kind of mistakes could lead to choosing each option. You must then provide your prediction as a letter by saying "Most Common Incorrect Answer:" at the end of your response.
>
> **Question:**
> This is the last day of 1899. What is the date tomorrow in MM/DD/YYYY?
>
> **Options:**

(A) 01/01/1900

(B) 01/22/1900

(C) 01/01/1899

(D) 02/06/1900

(E) 01/08/1900

(F) 01/01/1827

Remember that your task is NOT to give the correct answer, but to predict the most common incorrect answer.

**Anti-vote Self-Reflection Prompt:**

Anti-vote Self-Reflection Prompt

Your goal is to determine the most frequently proposed incorrect answer to a multiple-choice question among a set of solutions provided by other models. This task requires (i) analyzing the proposed solutions, (ii) identifying any logical fallacies or errors in each solution – potentially including the identification of the single correct answer, and (iii) deducing the most common incorrect answer from the provided distribution of predicted answers. Because there are many proposed solutions, you will be shown (i) the distribution of predicted answers (top-2), and (ii) one example response for each of those answers.

**Predicted answer distribution:**
Answer: B was given 14 times out of 20.
Answer: A was given 6 times out of 20.

**Example responses (one per answer above):**

**LLM 0 (Answer: B):** Let's carefully define the premises and break down the problem step-by-step to ensure accuracy and consistency with the given information. ### Premises - The day before yesterday was 11/23/1933. ### Step-by-Step Solution 1. **Identify "the day before yesterday,"** - According to the premise, the day before yesterday was 11/23/1933. 2. **Determine "yesterday,"** - If the day before yesterday was 11/23/1933, then yesterday was 11/24/1933. 3. **Determine "today,"** - If yesterday was 11/24/1933, then today must be 11/25/1933. 4. **Add 24 hours to today's date,** - Adding 24 hours to today's date (11/25/1933) brings us to 11/26/1933. 5. **Verify the given options,** - (A) 11/25/1933 - This is today, not the date 24 hours later. - (B) 11/26/1933 - This is 24 hours after today, 11/25/1933. - (C) 07/26/1934 - This is in a different month and year. - (D) 08/23/1933 - This is an earlier date. - (E) 11/26/1976 - This is a different year. - (F) 12/06/1933 - This is later in the year. ### Final Answer, Based on the detailed reasoning and the options provided, the correct date 24 hours after today (11/25/1933) is **11/26/1933**. Final Answer, B

**LLM 1 (Answer: A):** Let's carefully define the premises and break down the problem step-by-step to ensure accuracy and consistency with the given information. 1. **Identify the Given Date,** - The problem states, "The day before yesterday was 11/23/1933." 2. **Determine "Yesterday",** - If the day before yesterday was 11/23/1933, then yesterday must be 11/24/1933. 3. **Determine the Date 24 Hours Later,** - Adding 24 hours (which is one day) to 11/24/1933 will bring us to 11/25/1933. 4. **Verify the Given Options,** - (A) 11/25/1933 - (B) 11/26/1933 - (C) 07/26/1934 - (D) 08/23/1933 - (E) 11/26/1976 - (F) 12/06/1933 5. **Conclusion,** - The date 24 hours after 11/24/1933 is 11/25/1933, which matches option (A). Final Answer, A

Analyze different proposed solutions, aiming to identify any logical fallacies or mistakes made in each solution. After the analysis, you must provide your decision as a letter by saying "Most Common Incorrect Answer:" at the end of your response.

**Question:** The day before yesterday was 11/23/1933. What is the date 24 hours later in MM/DD/YYYY?

**Options:**

(A) 11/25/1933

(B) 11/26/1933

(C) 07/26/1934

(D) 08/23/1933

(E) 11/26/1976

(F) 12/06/1933

Remember that your task is NOT to give the correct answer, but to determine the most common incorrect answer.

