# OpenReview forum: "D-FUSEr: Diverse Failure, Unified Success via Error-Distribution Shaping in LLM Reasoning"
_ICML.cc/2026/Conference — ICML 2026 regular_

### Official Review · Reviewer_KDYJ · 2026-03-12

**Soundness:** 3
**Presentation:** 3
**Significance:** 2
**Originality:** 3
**Overall Recommendation:** 4
**Confidence:** 4

**Summary:**

This paper instroduces D-Fuser, a framework for improving LLM reasoning by modeling the distribution of erros. The authors argue that majorioty voting and rl training often fail as models repeatedly make the same wrong answer. They propose a diverse failure reward that encourages wrong answews to be less concentreated, and an inference time anti-vote mechanims that downweights the common mistake. The experiment shows that the proposed algorithm returns decent improvement.

**Compliance With Llm Reviewing Policy:**

Affirmed.

**Final Justification:**

Overall, the paper is well motivated. Originally, I had some concerns with the theoretical setups. However, it is fair for the authors to leave those for the future work. And I think the current contributions are decent enough for a conference paper.

**Key Questions For Authors:**

what would be some potential applications in some reasoning tasks, where there is no answer choices, like math or code training. Of cource, given the time constraint during the rebuttal, I don't request to see any actual experiemnts. But maybe some possible ideas would be great to know.

**Limitations:**

I don't see a limiation section. Having one in either main paper or appendix would be nice.

**Strengths And Weaknesses:**

strength:
- the problem that the authors trying solve is clear with well dicussed motivation.
- The method is supported with theoretical justifications
- Empirical results across different sizes and different family of open source model shows the improvement of the methods.

Weakenss:
- Some of the theories proposed relied on very strong assumptions such as a discrete category of error classes. However, in practical settings, these might not be true. Additional assumptions such as monotone updating, so it is also not fully clear how much of the theory carries over to more realistic open-ended answer spaces.
- Antivote would rely on the quality of antivote predictors, though this has been discussed by the authors in the paper.

---

> ### Author Rebuttal · Authors · 2026-03-30
>
> Thank you for your excellent comments! We are very glad to hear that you feel our approach is well motivated and validated via comprehensive experiments. Below, we have gone through each of your questions and comments; please let us know if anything is still unclear!
>
> > Some of the theories proposed relied on very strong assumptions such as a discrete category of error classes. However, in practical settings, these might not be true. Additional assumptions such as monotone updating, so it is also not fully clear how much of the theory carries over to more realistic open-ended answer spaces. (Weakness & Question)
>
> Thank you for raising this point. We agree that our framework, as presented, is most directly applicable to settings where answer equivalence can be readily assessed. However, we note that this limitation is not unique to D-FUSEr, this limitation is inherent to majority vote aggregation itself, which requires discrete, comparable answers to function. Majority vote is nonetheless one of the most widely studied and deployed test-time scaling methods, and improving its effectiveness in the settings where it applies is valuable in its own right.
>
> While beyond the scope of our work, we believe there are promising directions for extending our ideas beyond multiple-choice settings. For instance, one could use an LLM-as-judge or reward model to cluster open-ended outputs into equivalence classes, after which our error-distribution shaping techniques could be applied. We also note that the core insight, that correlated failures degrade aggregation, likely extends to other aggregation schemes (e.g., best-of-N with reward models), and exploring these connections is an exciting direction for future work. We will add a discussion of this limitation and these potential extensions to the paper.
>
>
> > Antivote would rely on the quality of antivote predictors, though this has been discussed by the authors in the paper.
>
> Thank you! Our goal was to show that training is not the only viable method for shaping error distributions. Anti-votes show promise as a means of shaping errors without training the model directly. Improving anti-vote efficacy could be done through a number of manners (training and training-free), We leave the problem of improving the antivote predictors for future work.
>
>
> We hope that our response has addressed your concerns and questions! If not, we are happy to provide further clarification. If all concerns have been adequately addressed, we kindly ask the reviewer to consider raising their score.

---

> > ### Author Rebuttal · Reviewer_KDYJ · 2026-04-02
> >
> > My concerns are resolved. I will keep my current score.

---

### Official Review · Reviewer_TwZs · 2026-03-13

**Soundness:** 3
**Presentation:** 3
**Significance:** 3
**Originality:** 3
**Overall Recommendation:** 4
**Confidence:** 4

**Summary:**

The paper proposes a method to favorably shift the error distribution of the trained policy to improve aggregation of sampled answers. Specifically, there are two contributions 1) a reward function encouraging answers to differ from the most common error in the prior and 2) a principled "anti-votes" strategy for self-consistency aggregation, where the models also predict the most common error to downvote. The authors test their approach on standard reasoning benchmarks, markedly improving performance over baselines.

**Compliance With Llm Reviewing Policy:**

Affirmed.

**Final Justification:**

The authors have addressed the main concerns I presented making the paper better in my opinion, however I do not think the work has enough significance to justify further increasing the score.

**Key Questions For Authors:**

1. I strongly encourage the authors to include a self-refine baseline where the policy is directly trained to optimize the majority vote score instead of pass@1.

**Limitations:**

The authors do not discuss the limitations of their proposed method.

**Strengths And Weaknesses:**

Strengths:
1. The approach is well motivated by the authors and fairly simple to implement.
2. The experiments empirically validate the claims, and I appreciate that the paper includes with a healthy variety of models, datasets, and baselines.
3. I find the anti-vote idea particularly interesting as a contribution, and the principled down-weighting proposed significantly improves performance.

Weaknesses:
1. Limited to tasks where solutions can be easily compared for equivalence (not true for code, proofs etc)
2. If the goal is to maximize majority voting score, a strong and simple alternative which the authors don't consider is to just do RL with the majority vote score of the group as the reward function.

---

> ### Author Rebuttal · Authors · 2026-03-30
>
> Thank you for your thoughtful and excellent comments! We are very glad to hear that you feel our approach is well motivated, simple to implement, and well validated via comprehensive experiments. Below, we have gone through each of your questions and comments; please let us know if anything is still unclear!
>
> > Limited to tasks where solutions can be easily compared for equivalence (not true for code, proofs etc)
>
> Thank you for raising this point. We discussed potential directions for extending our ideas beyond multiple-choice settings in the rebuttals for Reviewer KDYJ. In short, one could use an LLM-as-judge or reward model to cluster open-ended outputs into equivalence classes, after which our error-distribution shaping techniques could be applied.
>
> > If the goal is to maximize majority voting score, a strong alternative which the authors don't consider is to just do RL with the majority vote score..
>
> Thank you for the insightful suggestion. Directly optimizing the majority vote score was one of the first approaches we explored. Unfortunately, it performed worse using average accuracy, and we believe this is due to a fundamental credit assignment problem.
>
> **Why optimizing majority vote directly fails:** Consider computing the contribution of a single new answer $S_{\text{new}}$ given other answers $S_1, \ldots, S_M$ (with $M$ even for simplicity). The new answer can only change the majority vote outcome in two settings:
>
> 1.  Exactly $M/2$ of the existing answers are correct and $M/2$ are incorrect. Here, $S_{\text{new}}$ is the tiebreaker, if it is correct, the majority vote is correct, and vice versa.
>
>
> 2. Exactly $M/2 - 1$ answers fall on one side. Then $S_{\text{new}}$ can either create or break a tie.
>
>
> Crucially, both settings are unlikely events, they require the existing pool to be near-perfectly split. Moreover, even if we encounter the cases above, the reward reduces to rewarding individual correctness, since $S_{\text{new}}$ single-handedly dictates the outcome. The result is that directly optimizing majority vote accuracy collapses to a sparse version of an average accuracy reward.
>
> Alternatively, we could reward based on the number of correct answers in the majority. This would get rid of the sparsity issue, but we again recover something very similar to average accuracy.
>
> **Why use D-FUSEr:** This is precisely what motivated D-FUSEr: rather than trying to optimize the majority vote objective directly, we target the error distribution. This provides a denser training signal that is theoretically guaranteed to improve majority vote accuracy (Theorem 4.6). Designing a more sophisticated technique to make direct majority vote optimization viable is an interesting direction.
>
> We will add this discussion and the empirical comparison to the revised paper.
>
> Below we include a comparison of directly optimizing majority vote for Qwen-2.5-14B-Instruct: As predicted, Direct-Maj underperforms Dist-Reflect, confirming the credit assignment issue
> |                                   | OOD BBH | OOD GPQA | OOD MMPRO | OOD RBENCH |   ID BBH   |   ID MMPRO   |
> |--|---|--|---|---|---|--|
> | **Qwen-2.5-14B**   |         |       |       |         |         |              |
> | *— Untrained —* |         |          |           |            |            |              |
> | Zeroshot  | 87.7    | 44.4     | 68.2      | 58.4       | 87.7       | 68.2         |
> | Self-reflect    | 89.2    | 45.1     | 69.0      | 58.9       | 89.2       | 69.0         |
> | Dis-reflect | 87.2    | 43.9     | 65.3      | 56.1       | 87.2       | 65.3         |
> | *— Trained —* |         |          |           |            |            |              |
> | GRPO   | 85.2    | 46.7     | 64.7      | 51.7       | 89.9       | 69.5         |
> | Reflect-GRPO   | 84.3    | 43.3     | 65.3      | 52.5       | 90.1       | 69.2         |
> | Dist-Reflect + Direct-Maj (new)   | 87.1    | 44.9     | 66.2      | 57.8       | 87.0       | 67.2         |
> | Dist-Reflect  | 88.2    | **49.2** | 70.1      | 60.9       | **92.3**   | 69.3         |
> | D-FUSEr (ours)  | **91.9**| 48.9     | **73.5**  | **65.6**   | 91.4       | **75.7**     |
>
>
> > The authors do not discuss the limitations ...
>
> We will add the following to our limitations section to the camera-ready version:
>
> _The analysis relies on discrete answer spaces with well-defined answer extraction, making generalization to open-ended settings unclear. Distributional reflection introduces extra inference and training cost. Anti-votes require a sufficiently accurate predictor for full potential, which we leave for future work. For open-ended questions, clustering algorithms or LLM-based categorization could enable our framework, but this deserves separate investigation._
>
>
> We hope that our response has addressed your concerns and questions! If not, we are happy to provide further clarification. If all concerns have been adequately addressed, we kindly ask the reviewer to consider raising their score.

---

> > ### Author Rebuttal · Reviewer_TwZs · 2026-04-03
> >
> > I am overall satisfied with the response, but I think the score is appropriate so I will maintain it.

---

### Official Review · Reviewer_ME8f · 2026-03-15

**Soundness:** 3
**Presentation:** 3
**Significance:** 3
**Originality:** 3
**Overall Recommendation:** 4
**Confidence:** 3

**Summary:**

This paper studies how to improve majority voting and iterative refinement when LLM errors are highly concentrated. To address this, the paper proposes two methods: diverse failure reward, a theoretically motivated training objective that encourages more diverse failures, and anti-votes, an inference-time mechanism in which the model explicitly predicts likely incorrect answers. Empirically, across three model families and four benchmarks, the paper shows that both methods can improve majority-vote and iterative-refinement performance without hurting single-sample accuracy.

**Compliance With Llm Reviewing Policy:**

Affirmed.

**Key Questions For Authors:**

1. For the zero-shot inference baseline, is this direct inference or zero-shot CoT?

2. The paper should cite more related works showing that majority vote can be misleading. https://aclanthology.org/2025.findings-emnlp.660.pdf  also shows that majority vote fails because it ignores answer distribution structure and discussed the effect of iterative refinement. The authors should also cite the uncertainty quantification line of work, e.g., Semantic Entropy(https://arxiv.org/pdf/2302.09664). Even though it is framed as uncertainty estimation rather than vote aggregation, its core point is exactly that the distribution over semantically distinct answers matters more than raw surface counts.

**Limitations:**

See weaknesses and questions.

**Strengths And Weaknesses:**

Strengths:
1. The paper is well motivated by an interesting and important observation about language model error patterns. The proposed solutions are intuitive and supported by theory.

2. The paper is also clearly written overall, and the experiments cover a reasonably broad range of models and benchmarks.

Weaknesses:

My main concern is that some of the theoretical assumptions may be too restrictive or insufficiently justified.

1. The authors have assumption for fixed single-sample accuracy, the theory keeps the one-shot correctness probability fixed and studies only how redistributing the remaining error mass across wrong answers affects downstream voting or refinement. But in practice,  a method may change both how often the model is correct  and how its errors are distributed.

2. The strongest equivalence result is only shown for m=3 answers. The theory is narrower than the framing suggests as I understand the authors work on error-distribution shaping in general reasoning.

3. The iterative-refinement result depends on a monotone updating rule in definition 4.7, it’s unclear to me that whether real LLM refinement is well captured by that monotonicity notion and the authors need to further justify it.


On the experimental side, I have two concerns.

1. the paper should compare more explicitly against standard self-consistency / majority-vote baselines. While majority voting appears in parts of the evaluation, it should be treated as a primary baseline.

2. The anti-vote experiments are only conducted on a single dataset, which makes it hard to judge how broadly the results generalize.

---

> ### Author Rebuttal · Authors · 2026-03-30
>
> Thank you for your excellent questions and comments! We are very glad to hear that you feel our paper is clearly written with strong and comprehensive experimental results. Below, we have gone through each of your questions and comments; please let us know if anything is still unclear!
>
> > The authors assume fixed single-sample accuracy... in practice, a method may change both correctness and error distribution.
>
> Thank you for this thoughtful observation. You are correct that in practice, a method may change both single-sample accuracy and the error distribution simultaneously. The fixed-accuracy assumption is a deliberate theoretical choice to isolate the effect of error distribution on majority-vote accuracy, which is the core novel insight of our work. Without this controlled setup, it would be difficult to actually isolate the contributions of improved correctness and of error diversification on downstream correctness.
>
> Importantly, we fully account for this in practice, our total training reward is a composite of the correctness reward and the diverse failure reward (Section 3.3), so the model is explicitly incentivized to maintain or improve single-sample accuracy alongside diversifying its errors. The theory serves to motivate why error shaping helps (vs optimizing for correctness), by showing that, all else being equal, more uniform errors provably improves majority-vote accuracy, while the composite reward ensures that accuracy is not sacrificed in practice. We believe these two components are complementary: the theory isolates our proposed mechanism, and the empirical results (e.g., Tables 1 and 2) confirm that the combined objective improves both majority-vote accuracy and average accuracy across most settings, demonstrating that the two factors work together rather than trade off against each other.
>
> We will make this reasoning more explicit in the revised manuscript.
>
> > The strongest equivalence result is only shown for $m=3$ answers.
>
> We address this in detail in our response to Reviewer KZFQ. Briefly: if residual error mass satisfies $p_4 + \ldots + p_m \leq \varepsilon$, the equivalence holds up to error linear in $\varepsilon$. Moreover, Figure 3 empirically shows that error mass is heavily concentrated on the top-3 answers across all models and datasets, making $m=3$ a well-motivated and practically relevant regime.
>
> > The iterative-refinement result depends on a monotone updating rule... unclear whether real LLM refinement is well captured by that notion.
>
> We also address this in our response to Reviewer KZFQ. In short, Figure 13 in the supplement shows that LLM error concentration increases between rounds, confirming approximate monotonicity in practice. Prior work further corroborates this, e.g., that as answer frequency increases, so too does the probability that an LLM will repeat that answer at the next round.
>
> > the paper should compare more explicitly against standard self-consistency / majority-vote baselines
>
>
> Thank you for the comment! In Table 1, we showed direct comparisons between our method and standard majority-vote baselines.
>
> > The anti-vote experiments are only conducted on a single dataset.
>
> Thank you for the comment! We had included our anti-vote experiment results on other datasets (BBH, GPQA, MMPRO, RBENCH) in Appendix D, Figure 16 of our original submission.
>
> > For the zero-shot inference baseline, is this direct inference or zero-shot CoT?
>
> Our zero-shot inference baseline is zero-shot CoT; we included an example of our zero-shot prompts in Appendix E of our original submission.
>
> > The paper should cite more related works... majority vote can be misleading... Semantic Entropy…
>
>
> Thank you for bringing these related works to our attention! We will cite them in our camera-ready version (We are not allowed to update our submission during the discussion period).
>
>
> We hope that our response has addressed your concerns and questions! If not, we are happy to provide further clarification. If all concerns have been adequately addressed, we kindly ask the reviewer to consider raising their score.

---

> > ### Author Rebuttal · Reviewer_ME8f · 2026-04-03
> >
> > I am satisfied with the authors' response and I will keep my current score.

---

### Official Review · Reviewer_KZFQ · 2026-03-16

**Soundness:** 3
**Presentation:** 3
**Significance:** 3
**Originality:** 3
**Overall Recommendation:** 4
**Confidence:** 4

**Summary:**

This paper proposes D-FUSEr, a framework designed to imrove inference time scaling methods such as majority vote aggregation and iterative refinement for large language model reasoning. The central premise is that the effectiveness of aggregation methods depends not only on the correctness of individual samples but also on the distribution of errors across those samples. The authors argue that when model errors are concentrated on a few dominant incorrect answers, majority vote and refinement strategies may reinforce these shared mistakes rather than crrect them.

To address this issue, the paper introduces two mechanisms aimed at shaping the distribution of model errors. First, the authors propose a diverse failure reward, a training objective that encourages incorrect predictions to differ from the most common mistake. This objective is theoretically motivated and shown to improve expected majority vote accuracy under certain assumptions about error distributions. Second, the paper proposes anti-votes, an inference time mechanism in which the model predicts a likely incorrect answer alongside its predicted solution, allowing aggregation methods to down weight dominant errors.

The paper also introduces distributional reflection, a refinement strategy where the model conditions on the empirical distribution of previous solutions rather than a single prior response. The approach is evaluated across several open source LLMs and reasoning benchmarks including BBH, MMLU-Pro, GPQA, and RBENCH. The results show improvements in majority vote performance and some gains in average accuracy when using the proposed training objective and anti-vote mechanism.

**Compliance With Llm Reviewing Policy:**

Affirmed.

**Key Questions For Authors:**

Q1. Anti-votes improve results, but the paper leaves anti-vote predictor training to future work. In the current experiments, how are anti-votes elicited exactly, and what is their accuracy relative to predicting the dominant mistake? A clearer answer would help determine whether this is a practical method or mainly an oracle-like analysis tool.

Q2. The paper focuses heavily on majority-vote accuracy with 20 samples. How sensitive are the gains to the number of samples, and do the benefits persist when inference budgets are smaller or larger? This matters for practical significance.

**Limitations:**

The paper includes a brief impact statement and acknowledges that decorrelating errors can increase noisiness and does not eliminate mistakes, but the discussion of limitations is still too thin. It should more explicitly discuss the gap between the theory and realistic multi-answer or open-ended settings, the dependence on answer extraction into discrete classes, the added inference and training cost of distributional reflection, the incomplete practicality of anti-votes without a trained predictor, and the fact that gains are concentrated more strongly in majority-vote settings than in single-sample or average-accuracy settings.

**Strengths And Weaknesses:**

The paper has a clear and interesting core insight. Much of the recent test-time scaling literature treats samples as interchangeable and focuses on better search, more samples, or stronger self-correction. This submission instead asks whether the geometry of model failures itself should be optimized. That is a meaingful reframing, and I do think the paper identifies a real failure mode of majority vote and refinement when errors are correlated. The empirical observation that error mass is often concentrated on a few dominant wrong answers is plausible and directly relevant to practice. The combination of theory plus experiments also makes the paper more compelling than a purely empirical proposal.

That said, the paper overstates the breadth of its theoretical contribution relative to what is actually proved. The cleanest equivalence result between the proposed reward and majority-vote accuracy is only established for the case m = 3, and the iterative-refinement result depends on an abstract monotone updating rule rather than a concrete characterization of real LLM refinement dynamics. The paper argues this restriction is motivated by empirical concentration on top answers, but that does not fully close the gap between the theorem and the real benchmark settings, many of which have far more than three candidate answers or open-ended reasoning spaces before answer extraction. In other words, the theory is directionally suggestive, but not yet strong enough to justify the broader claim that the reward directly optimizes majority-vote performance in realistic settings.

On the experimental side, the results are promising but not fully disentangled. A major issue is that D-FUSEr combines at least two conceptually distinct changes, namely distributional reflection and the diverse failure reward, and the paper itself notes that distributional reflection alone can already outperform GRPO and Reflect-GRPO in several settings. In Table 1, D-FUSEr often wins on majority-vote accuracy, but Dist-Reflect is already quite strong, and in some cases the margins between Dist-Reflect and D-FUSEr are modest or inconsistent across models and datasets. Likewise, Table 2 shows that average accuracy improvements are more mixed, which weakens the claim that the method broadly improves reasoning rather than primarily helping an aggregation-specific metric. What is missing is a sharper ablation that isolates how much of the gain comes from the reward versus the richer conditioning context versus the specific RL setup.

I also found the anti vote story somewhat incomplete. The idea is intuitive and the empirical gains in Figure 6 are notable, but the paper explicitly acknowledges that it does not solve the problem of training an accurate anti vote predictor. As presented, anti votes are closer to an interesting auxiliary signal than a mature method. Relatedly, the dynamic anti vote weighting uses a theoretically derived formula that may not be easily operationalized in practice, which the paper also concedes. This makes the anti-vote component feel more like a proof of concept than a finished contribution. The same concern applies to the "optimality gap" analysis, which is built by uniformly redistributing error mass over the top-4 wrong answers. That diagnostic is reasonable as an illustrative lens, but it is not obvious that this construction corresponds to an achievable policy class for real models, so I would treat it as suggestive rather than strong evidence.

In terms of originality and significance, I would place the paper in a middle ground. The high level components are not individually new: self-reflection, multisample conditioning, reward shaping, and reweighting of votes all have strong precedents. The contribution is in the way they are unified around the notion of error-distribution shaping and tied to majority vote behavior. That is a useful perspective, and I can imagine follow up work building on it. However, I am less convinced that the current paper fully establishes a broadly reusable principle for the community, rather than a somewhat specialized improvement targeted at aggregation-heavy multiple-choice reasoning pipelines. My overall take is that the paper is more than incremental in framing, but still somewhat incremental in mechanism.

---

> ### Author Rebuttal · Authors · 2026-03-30
>
> Thank you for your insightful questions and comments! We are glad you find our paper has a clear insight with strong empirical results. We address each point below.
>
> > the paper overstates the breadth of its theoretical contribution... relies on m=3...monotone updating rule…
>
> **m=3** While $m=3$ yields the cleanest results, it also provides the design principle behind our reward. If residual error mass satisfies $p_4 + \ldots + p_m \leq \varepsilon$, the equivalence between D-FUSEr reward and majority-vote accuracy holds up to a small error linear in $\varepsilon$. To see this consider the case with $4$ options (the same argument holds for any number), suppose $p_2 > p_3 > p_4 = \varepsilon$, and after training, $p_2' = p_2 - x, p_3' = p_3 + x + \varepsilon, p_4' = 0$ (worst case mass shift). Since $p_2' < p_2$, the D-FUSEr reward increases. However, since $p_3’ > p_3$, majority vote accuracy could be worse by a factor of $\varepsilon$. Importantly, this setting is rare in practice; continued training with D-FUSEr will decrease the mass of $p_3’$ further. $\varepsilon$ is quite small in nearly all cases (Figure 3).
>
> Note that our empirical gains confirm the mechanism extends beyond $m=3$.
>
> **Monotonicity** We observe that LLMs roughly follow this rule in (Figure 13 error concentration from one round to the next). Prior work corroborates this, e.g., “Multi-LLM Debate: Framework, Principals, and Interventions” shows majority sizes grow monotonically across rounds.
>
> > results are not fully disentangled… distributional reflection already outperforms GRPO
>
> We clarify that our target metric is **majority-vote accuracy**, not average accuracy. Table 2 (avg accuracy) is provided to show D-FUSEr does not degrade single-sample performance.
>
> Ablations to help isolate the effects of D-FUSEr are present in our results:
>
> - **GRPO vs Dist-Reflect (trained)** Shows the isolated effect of distributional reflection with a standard reward.
>
> - **Dist-Reflect (trained) vs D-FUSEr** Shows the isolated effect of the diverse failure reward. D-FUSEr surpasses Dist-Reflect beyond the confidence interval in **15/18** (model, dataset) pairs.
>
> - **Mechanism** Figures 4,15 reveals why D-FUSEr is effective; Dist-Reflect + GRPO often _worsens_ error concentration (echo-chamber effect). D-FUSEr specifically counteracts this, consistently achieving the lowest error concentration.
>
> Our average accuracy gains are a secondary benefit: by increasing answer diversity, D-FUSEr reduces homogeneous training groups where all advantages are 0, allowing the underlying GRPO framework to better reinforce correct paths.
>
> > anti-vote story is incomplete… How are anti-votes elicited?
>
> Anti-votes are elicited by prompting the model to predict the most likely incorrect answer. We will include the full prompt in the camera-ready. Anti-vote accuracy for predicting the dominant mistake:
>
> ||Qwen3B|Qwen7B|Qwen14B|Gemma|Llama|
> |-|-:|-:|-:|-:|-:|
> |BBH|32.6|26.5|26.0|21.8|38.9|
> |GPQA|31.0|26.1|28.8|17.4|30.4|
> |MMPRO|58.0|57.1|18.2|50.9|62.1|
> |RBENCH|39.3|61.7|35.3|85.8|47.9|
>
> Our goal is to demonstrate anti-votes as a promising paradigm; training an accurate predictor is an exciting direction for future work.
>
> **Optimality gap clarification** The gap is a diagnostic, not a claim about achievable policies. D-FUSEr consistently achieves the smallest gap (Figures 5,14), confirming its reward shapes errors toward the theoretical optimum.
>
> > paper is more than incremental in framing, but still somewhat incremental in mechanism.
>
> We agree individual components are not new in isolation. Our contribution is identifying **error distribution shaping** as a distinct optimization target, and providing formal guarantees (Theorems 4.5, 4.6, 4.8). To our knowledge, no prior work directly shapes error distributions through training to improve aggregation and generally lacks theoretical grounding. This perspective can extend beyond our focus to any multi-output setting (ensembles, best-of-N, multi-agent systems, etc.).
>
> > How sensitive ... to the number of samples?
>
> We chose $n=20$ as sufficient to approximate the underlying distribution. In initial experiments we saw the majority vote of all methods plateau around $n=20$. For very small $n$ (2-3), gains remain to be evaluated.
>
> > limitations should more explicitly discuss the gap between theory and realistic settings…
>
> We will add the following to the camera-ready version:
>
> _The analysis relies on discrete answer spaces with well-defined answer extraction, making generalization to open-ended settings unclear. Distributional reflection introduces extra inference and training cost. Anti-votes require a sufficiently accurate predictor for full potential, which we leave for future work. For open-ended questions, clustering algorithms or LLM-based categorization could enable our framework, but this deserves separate investigation._
>
> We hope our responses address your concerns. If so, we kindly ask you to consider raising your score.

---

> > ### Author Rebuttal · Reviewer_KZFQ · 2026-04-05
> >
> > I acknowledge and thank the writers for providing the detailed rebuttal, but I still stand with my previous rating.

---

### Decision · Program_Chairs · 2026-04-30

**Decision:**

Accept (regular)

**Comment:**

This paper proposes to improve LLM reasoning by shaping the distribution of model errors so that aggregation methods work better. Reviewers were uniformly positive and agreed that the central insight is timely and practically meaningful: aggregation quality depends not only on per-sample accuracy, but also on how errors are distributed across samples. Both the diverse-failure objective and the anti-vote mechanism were viewed as conceptually appealing, and reviewers found the empirical results across several reasoning benchmarks and model families convincing.

The remaining concerns were secondary. Reviewers asked for a fuller limitations discussion, clearer treatment of the assumptions connecting the theory to realistic open-ended reasoning settings, and more explanation of the practicality of anti-votes without dedicated predictors. I read the authors' discussion responses, which appear constructive and responsive to these points. In my view, these are revision requests rather than barriers to acceptance. The paper offers a clear idea, strong empirical support, and useful lessons for LLM reasoning, so I recommend acceptance.